# Feature-Aware (Hyper)graph Generation via Next-Scale Prediction

**Dorian Gailhard** [1]  **Enzo Tartaglione** [1]  **Lirida Naviner** [1]  **Jhony H. Giraldo** [1]

## Abstract

Graph generative models perform well on small structured data but struggle to scale to large, complex structures. Hierarchical approaches improve scalability but often ignore node and edge features, which are critical in real-world applications, particularly for hypergraphs that model higher-order relationships. In this paper, we propose FAHNES (**f**eature-**a**ware (**h**yper)graph generation via **ne**xt-**s**cale prediction), a hierarchical framework that jointly generates topology and features for graphs and hypergraphs. FAHNES builds multi-scale representations through node coarsening and localized expansion, guided by a novel hierarchical scale encoding that controls granularity and ensures cross-scale consistency. Experiments on synthetic, 3D mesh, and graph point cloud datasets demonstrate competitive or state-of-the-art performance while uniquely scaling to featured large-scale graphs and hypergraphs. Our code is open source[1].

## 1. Introduction

Generating discrete geometric structures such as graphs and hypergraphs is an important challenge in machine learning (Zhu et al., 2022). These structures are central to applications ranging from molecular design and materials discovery to electronic circuits and 3D shape modeling (Kajino, 2019; Rahman et al., 2012; Luo et al., 2024). Their ability to capture complex relationships—pairwise in graphs, and multi-way in hypergraphs—makes them an essential tool for representing and synthesizing structured data. In many real-world settings, topology alone is insufficient: node and edge (or hyperedge) features often carry important semantic

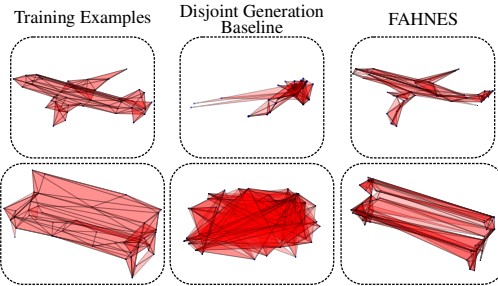

*Figure 1.* Examples of generated featured hypergraphs by a sequential disjoint generation baseline and our model (FAHNES).

or geometric information, such as coordinates in 3D meshes.

Despite recent advances, existing methods for featured graph generation struggle to scale. Most of these approaches use flat architectures that model the entire structure at once, leading to quadratic computational and memory complexities (Vignac et al., 2023; Eijkelboom et al., 2024; Xu et al., 2024). This limits their use to moderately sized graphs. To address scalability, hierarchical generation strategies have been proposed for both graphs and hypergraphs (Bergmeister et al., 2024; Gailhard et al., 2025). These methods build the structure in stages, starting from a coarse representation and progressively upsampling and refining it, which allows them to handle much larger topologies.

However, existing hierarchical methods only focus on unfeatured structures (Bergmeister et al., 2024; Gailhard et al., 2025). Extending them to generate features is challenging because different regions of a graph or hypergraph often grow at uneven rates during the refinement process, making it difficult to maintain consistency across scales. Furthermore, sequentially generating topology first and then features—rather than modeling them jointly—is ineffective in complex settings, as illustrated in Fig. 1.

To overcome these limitations, we propose FAHNES (**f**eature-**a**ware (**h**yper)graph generation via **ne**xt-**s**cale prediction), a hierarchical generative framework that jointly models topology and features for both graphs and hypergraphs.[2] FAHNES builds multi-scale representations (Tian et al., 2024) through node coarsening

[1]LTCI, Télécom Paris, Institut Polytechnique de Paris, France. Correspondence to: Dorian Gailhard <dorian.gailhard@telecom-paris.fr>, Enzo Tartaglione <enzo.tartaglione@telecom-paris.fr>, Lirida Naviner <lirida.naviner@telecom-paris.fr>, Jhony H. Giraldo <jhony.giraldo@telecom-paris.fr>.

*Proceedings of the 43rd International Conference on Machine Learning*, Seoul, South Korea. PMLR 306, 2026. Copyright 2026 by the author(s).

[1]https://github.com/DorianGailhard/FAHNES

---

[2]As hypergraphs generalize graphs, we present FAHNES for hypergraphs; the methodology is then easily adapted for graphs.

and reconstructs fine structures via localized expansion, while directly predicting features alongside structure at each stage. A fundamental component of our model is a *hierarchical scale encoding*, which encodes local growth constraints and helps maintain consistency in regions that expand at different rates. In addition, we propose a *multi-scale graph optimal-transport (OT) coupling*, which generalizes the mini-batch OT coupling (Tong et al., 2024; Pooladian et al., 2023) to our hierarchical graph setting, improving stability and coherence during generation. This combination enables scalable generation of large, featured graphs or hypergraphs across diverse domains, from 3D meshes to point clouds. Our main contributions are:

- We introduce the first hierarchical generative method, both for graphs and hypergraphs, able to generate topology and features, and having quasi-linear complexity (Section 3.1).
- We propose a novel hierarchical scale encoding that enhances global structural coherence in hierarchical graph and hypergraph generation (Sections 3.2 and 3.3).
- To address the permutation-alignment problem in graphs, we propose a theoretically grounded multi-scale OT coupling for graph and hypergraph generation (Section 3.5).
- We validate FAHNES on both synthetic and real-world datasets, demonstrating strong performance in jointly generating topology and features (Section 5).

## 2. Related Work

**Graph and hypergraph generation using deep learning**. Graph generation has seen significant progress in recent years. Early approaches, such as GraphVAE (Simonovsky & Komodakis, 2018), used autoencoders to embed graphs into latent spaces for sampling. Subsequent models employed recurrent neural networks to sequentially generate adjacency matrices, improving structural fidelity (You et al., 2018). More recently, diffusion-based methods enabled permutation-invariant graph generation (Niu et al., 2020; Vignac et al., 2023), with extensions incorporating structural priors such as node degrees (Chen et al., 2023). Many prior methods jointly model node features and topology, but remain limited to small graphs due to scalability issues. In particular, diffusion-based models (Vignac et al., 2023; Eijkelboom et al., 2024; Xu et al., 2024) operate on complete graphs by corrupting structures and features and learning to denoise them, which limits scalability due to the combinatorial number of possible edges.

To mitigate this, hierarchical methods have been proposed. Bergmeister et al. (2024) introduced a scalable graph generation framework based on a coarsen-then-expand approach, merging nodes to form coarse representations and progressively reconstructing finer details. This framework was extended to hypergraphs by Gailhard et al. (2025), which allows edges to connect more than two nodes. However, both methods only focus on topology generation, neglecting node and hyperedge features essential for many applications.

Hierarchical graph generation has been explored extensively in molecular modeling (Qiang et al., 2023; Kuznetsov & Polykovskiy, 2021; Zhu et al., 2023). However, these methods are highly tailored to the molecular domain and are difficult to extend to large graphs. In contrast, our approach jointly models features and connectivity at all scales and is not restricted to molecular generation, enabling broader and more scalable hierarchical modeling.

**Applications of hypergraph generation**. Generative models that capture higher-order structure and node or hyperedge features are crucial in many domains. For example, 3D shape modeling involves surfaces (*e.g.*, triangles, quads, or polygons) that go beyond pairwise connectivity. Existing methods often rely on fixed topology, quantization, or autoregressive transformer-based modeling (Nash et al., 2020; Siddiqui et al., 2024), limiting flexibility and scalability. Hypergraphs offer a general representation by treating faces as hyperedges, enabling joint topology and feature generation.

Unlike prior work that focuses solely on topology or relies on flat, non-scalable designs, we present the first unified, scalable approach for jointly modeling topology and features for graphs and hypergraphs. FAHNES enhances hierarchical generative modeling with feature integration, hierarchical scale encoding, and multi-scale graph OT coupling.

## 3. Feature-Aware (Hyper)graph Generation via Next-Scale Prediction

**Notations**. We use calligraphic letters (*e.g.*, $\mathcal{V}$) for sets, with cardinality $|\mathcal{V}|$. Bold uppercase letters denote matrices (*e.g.*, $\mathbf{A}$), and bold lowercase letters denote vectors (*e.g.*, $\mathbf{x}$). The transpose and element-wise multiplication are written $(\cdot)^{\top}$ and $\odot$, respectively. $\lceil \cdot \rceil$ denotes rounding to the nearest integer, and $Id(x) = x$ is the identity function.

**Basic definitions**. We define a graph $G = (\mathcal{V}, \mathcal{E})$ as a pair consisting of a set of vertices $\mathcal{V}$ and a set of edges $\mathcal{E} \subseteq \mathcal{V} \times \mathcal{V}$. Graphs may also carry node and edge features, represented by matrices $\mathbf{F}_{\mathcal{V}} \in \mathbb{R}^{|\mathcal{V}| \times m}$ and $\mathbf{F}_{\mathcal{E}} \in \mathbb{R}^{|\mathcal{E}| \times l}$, where $m$ and $l$ denote the dimensionality of the features. Each edge $e \in \mathcal{E}$ corresponds to a pair $(u, v)$, indicating a connection between nodes $u$ and $v$. A bipartite graph $B = (\mathcal{V}_L, \mathcal{V}_R, \mathcal{E})$ is a special case of a graph where the vertex set is split into two disjoint subsets $\mathcal{V}_L$ and $\mathcal{V}_R$, and edges exist only between the two parts, *i.e.*, $\mathcal{E} \subseteq \mathcal{V}_L \times \mathcal{V}_R$. The full set of nodes is $\mathcal{V} = \mathcal{V}_L \cup \mathcal{V}_R$. In this work, we consider node features for bipartite graphs, denoted by $\mathbf{F}_L$ for the left-side nodes and $\mathbf{F}_R$ for the right-side nodes.

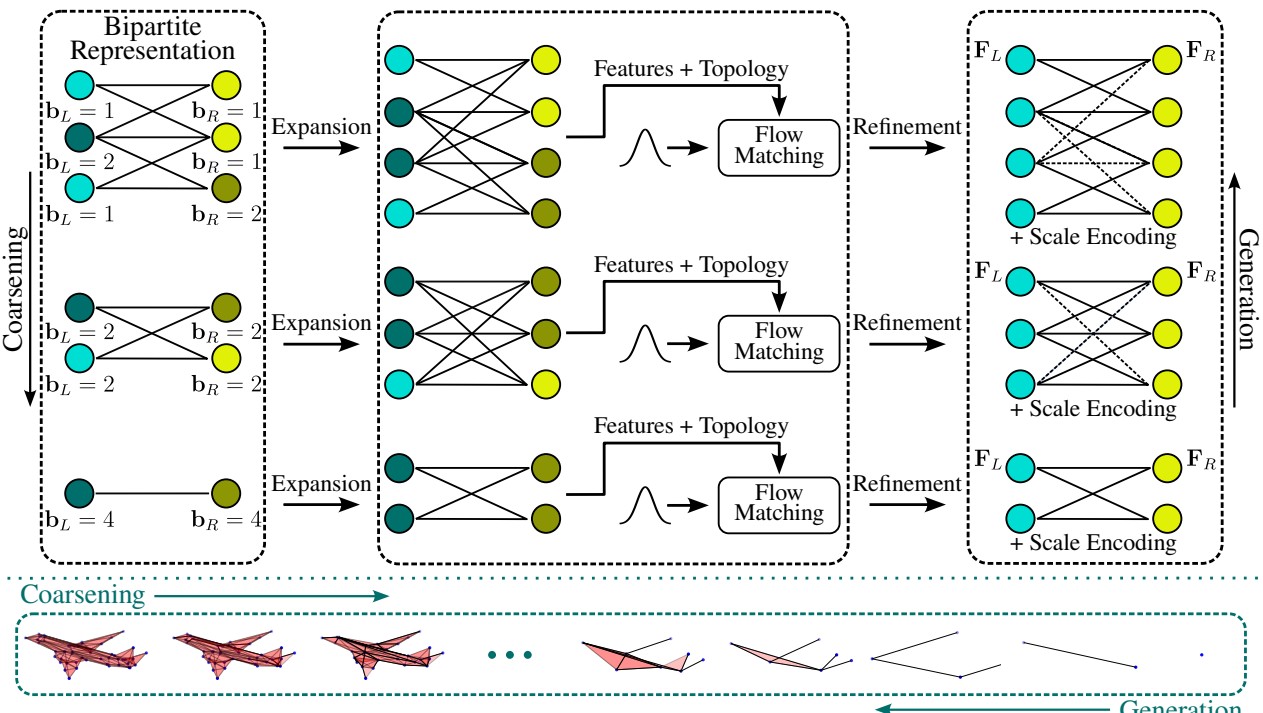

*Figure 2.* Our framework adopts a *coarsening-expansion* strategy. *i*) During training, input hypergraphs are progressively coarsened by merging nodes and hyperedges, yielding a multiscale representation. Node features are averaged during merging, and scale encodings are summed. *ii*) The model is then trained to predict which nodes were merged at each scale. *iii*) In the *expansion* phase, merged nodes (shown in dark in the leftmost column) are expanded back (copies shown in dark), inheriting their parent's features, scale encoding, and connectivity. In the *refinement* phase, the model is trained to (*a*) identify which edges should be removed (dotted lines), (*b*) predict the scale encodings of the children based on the parent's scale encoding, and (*c*) refine the features of newly expanded nodes.

A hypergraph $H$ extends the concept of a graph and is specified by a pair $(\mathcal{V}, \mathcal{E})$, where $\mathcal{V}$ represents the vertex set and $\mathcal{E}$ comprises hyperedges, with each $e \in \mathcal{E}$ being a subset of $\mathcal{V}$. The main distinction of hypergraphs lies in their ability to connect arbitrary numbers of vertices through a single hyperedge. Similar to graphs, hypergraphs can possess node and hyperedge features, which are denoted $\mathbf{F}_{\mathcal{V}}$ and $\mathbf{F}_{\mathcal{E}}$. We consider two fundamental graph-based representations of hypergraphs: clique and star expansions. The clique expansion transforms a hypergraph $H$ into a graph $C = (\mathcal{V}, \mathcal{E}_c)$, where hyperedges are replaced by cliques, *i.e.*, $\mathcal{E}_c = \{(u,v) \mid \exists\, e \in \mathcal{E} : u, v \in e\}$. The star expansion converts a hypergraph $H$ into a bipartite structure $B = (\mathcal{V}_L, \mathcal{V}_R, \mathcal{E}_b)$, where $\mathcal{V}_L = \mathcal{V}$, $\mathcal{V}_R = \mathcal{E}$, and $\mathcal{E}_b = \{(v, e) \mid v \in \mathcal{V}_L, e \in \mathcal{V}_R, v \in e \text{ in } H\}$. Left side nodes $\mathcal{V}_L$ represent the nodes of the hypergraph, while right side nodes $\mathcal{V}_R$ represent hyperedges.

Our objective is to develop a generative model that can sample from the underlying distribution of a dataset of featured hypergraphs (or graphs) $(H_1, \ldots, H_N)$, *i.e.*, to learn the joint distribution over both topology and features. We present FAHNES in the more general setting of hypergraphs; the method can be trivially adapted to graphs, as discussed in Section 3.6. Complete mathematical proofs of all propo-

sitions in this paper appear in Appendix B.

### 3.1. Overview

FAHNES follows a hierarchical pipeline (Fig. 2): *i*) during training, multi-scale representations of input (hyper)graphs are constructed; *ii*) a model learns to reconstruct higher scales from lower ones. We use a coarsening–expansion framework (Gailhard et al., 2025). Multi-scale representations are obtained via spectrum-preserving coarsening (Loukas, 2019) on the clique expansion of the hypergraph. The process is then reversed through learned expansion and refinement, reconstructing topology and features at progressively finer scales. Expansion and refinement are learned on the bipartite graph using flow matching (Lipman et al., 2023), treating generation as the inverse of coarsening.

To address uneven growth across regions, we introduce a *hierarchical scale encoding*: each cluster is assigned a scale encoding indicating its remaining expansions. Scale encodings are recursively divided among child nodes, starting from a single super-node with the full scale encoding and ending when all clusters have a scale encoding of one. This provides more local control over the final node count and improves global consistency compared to prior methods

(Bergmeister et al., 2024; Gailhard et al., 2025), which only append the desired size to node embeddings. At each expansion step, the model also predicts the mean feature of future child nodes conditioned on the parent's feature, extending the scale-wise autoregressive idea in (Ren et al., 2025) so that predictions at one scale guide those at the next.

## 3.2. Coarsening with Hierarchical Scale Encoding

We adopt the coarsening strategy in (Gailhard et al., 2025), representing hypergraphs at multiple resolutions by merging nodes into clusters. Node pairs are merged via spectrum-preserving coarsening (Loukas, 2019) applied to the hypergraph's clique expansion. These mergings are mirrored in the bipartite representation by merging left-side nodes and collapsing duplicate hyperedges. We restrict node mergings to at most two per cluster, which bounds hyperedge mergings to at most three (Gailhard et al., 2025). For each resulting cluster, we track two quantities. First, the *scale encoding* equals the number of nodes in the cluster: it is initialized to 1 for each node and hyperedge, summed upon merging, and assigned to the resulting super-cluster. This encoding constrains local growth by specifying how many fine-level nodes a coarse cluster should expand into, guiding refinement across regions with different growth rates. Second, the super-cluster feature is computed as the weighted mean of its constituent features, representing the average feature of the contained nodes.

We denote node (left-side) scale encodings and features by $\mathbf{b}_L \in \mathbb{N}^{|\mathcal{V}_L|}$ and $\mathbf{F}_L$, and hyperedge (right-side) scale encodings and features by $\mathbf{b}_R \in \mathbb{N}^{|\mathcal{V}_R|}$ and $\mathbf{F}_R$.

**Definition 1** (Bipartite graph coarsening). Let $H$ be a hypergraph, $C = (\mathcal{V}_c, \mathcal{E}_c)$ its clique expansion, and $B = (\mathcal{V}_L, \mathcal{V}_R, \mathcal{E}_b, \mathbf{b}_L, \mathbf{b}_R, \mathbf{F}_L, \mathbf{F}_R)$ its *featured* bipartite representation. Let $\mathcal{P}_L = \{\mathcal{V}^{(1)}, \ldots, \mathcal{V}^{(n)}\}$ be a partitioning[3] of the node set $\mathcal{V}_L$ such that each set $\mathcal{V}^{(p)}$ induces a connected subgraph in $C$. We construct an intermediate coarsening $\tilde{B}(B, \mathcal{P}_L) = (\bar{\mathcal{V}}_L, \mathcal{V}_R, \bar{\mathcal{E}}_b, \bar{\mathbf{b}}_L, \mathbf{b}_R, \bar{\mathbf{F}}_L, \mathbf{F}_R)$ by merging each $\mathcal{V}^{(p)}$ into a single node $v^{(p)} \in \bar{\mathcal{V}}_L$, and by defining:

$$\bar{\mathbf{b}}_L[p] = \sum_{v \in \mathcal{V}^{(p)}} \mathbf{b}_L[v], \ \bar{\mathbf{F}}_L[p] = \frac{1}{\bar{\mathbf{b}}_L[p]} \sum_{v \in \mathcal{V}^{(p)}} \mathbf{b}_L[v]\mathbf{F}_L[v], \tag{1}$$

for every merged node $\mathcal{V}^{(p)}$. An edge $e_{\{p,q\}} \in \bar{\mathcal{E}}_b$ is added between $v^{(p)} \in \bar{\mathcal{V}}_L$ and $v^{(q)} \in \mathcal{V}_R$ if there exists an edge $e_{\{i,q\}} \in \mathcal{E}_b$ in the original bipartite representation between some $v^{(i)} \in \mathcal{V}^{(p)}$ and $v^{(q)}$.

To complete the coarsening process, we define an equivalence relation $v_1 \sim v_2 \iff \mathcal{N}(v_1) = \mathcal{N}(v_2)$ on $\mathcal{V}_R$, where $\mathcal{N}(v)$ denotes the set of neighbors of $v$, *i.e.*, we con-

sider right side nodes having the same set of neighbors as equivalent, or in other words we consider hyperedges containing the same set of nodes as equivalent. This induces a partitioning $\mathcal{P}_R = \{\mathcal{V}_R^{(1)}, \ldots, \mathcal{V}_R^{(m)}\}$, allowing us to construct the fully coarsened bipartite representation $\bar{B}(\tilde{B}, \mathcal{P}_L) = (\bar{\mathcal{V}}_L, \bar{\mathcal{V}}_R, \bar{\mathcal{E}}_b, \bar{\mathbf{b}}_L, \bar{\mathbf{b}}_R, \bar{\mathbf{F}}_L, \bar{\mathbf{F}}_R)$ by merging each part $\mathcal{V}_R^{(p)}$ into a single node $v_R^{(p)} \in \bar{\mathcal{V}}_R$, similarly to the construction of $\bar{\mathcal{V}}_L$, and by defining:

$$\bar{\mathbf{b}}_R[p] = \sum_{v \in \mathcal{V}_R^{(p)}} \mathbf{b}_R[v], \ \bar{\mathbf{F}}_R[p] = \frac{1}{\bar{\mathbf{b}}_R[p]} \sum_{v \in \mathcal{V}_R^{(p)}} \mathbf{b}_R[v]\mathbf{F}_R[v], \tag{2}$$

for every merged hyperedge $\mathcal{V}_R^{(p)}$.

*Remark* 1. Informally, we cluster nodes in the clique expansion, merge the corresponding left-side nodes of the bipartite graph, and compute their scale encodings and features as described above. Right-side nodes (hyperedges) connected to the same left-side nodes are then merged. In our implementation, we select nodes for clustering using spectrum-preserving coarsening (Loukas, 2019). In the final reduction—when only one node and one hyperedge remain—we replace their features with zero matrices to yield a feature-agnostic initialization for generation. Please note that cluster size vectors introduced in (Bergmeister et al., 2024) correspond to the sizes of all sets $\mathcal{V}_L^{(p)}$ and $\mathcal{V}_R^{(p)}$, *i.e.*, the number of mergings for each cluster at this specific step, and not the number of nodes absorbed since the initialization of coarsening—which we call *scale encoding*. Further details about the coarsening methodology are provided in Appendix C. Some visual examples of coarsening sequences are provided in Appendix D.

We provide the following theoretical result regarding the feature merging process in FAHNES.

**Proposition 1.** *Let* $\mathbf{X}$ *be the node feature matrix and* $\mathcal{P} = \{\mathcal{C}_k\}_{k=1}^K$ *a partition of the vertices into clusters. The optimal downsampled features minimizing the reconstruction error are the cluster-wise barycenter:*

$$\mathbf{x}_{\mathcal{C}_k}^* = \frac{1}{|\mathcal{C}_k|} \sum_{v \in \mathcal{C}_k} \mathbf{x}_v \quad for \ 1 \leq k \leq K. \tag{3}$$

## 3.3. Expansion and Refinement with Hierarchical Scale Encoding

Once multi-scale representations are obtained via coarsening, the objective is to train a model to reverse this process. We define the inverse of coarsening, which the model learns to imitate, consisting of two stages: expansion (upsampling by duplicating nodes and hyperedges) and refinement (adjusting connectivity, scale encodings, and features). This process operates exclusively on the bipartite representation.

Since our framework only conditions on the total number of nodes in the final hypergraph, we maintain a single node

---

[3]That is, $\mathcal{V}^{(p)} \subseteq \mathcal{V}_L$, $\bigcup_{i=1}^n \mathcal{V}^{(i)} = \mathcal{V}_L$, and $\mathcal{V}^{(i)} \cap \mathcal{V}^{(j)} = \emptyset \ \forall \ 1 \leq i, j \leq n$.

scale encoding vector $\mathbf{b} \in \mathbb{N}^{|\mathcal{V}_L|}$ for the node (left-side) partition and discard the hyperedge (right-side) scale encoding. To undo mergings, clusters are recursively split into multiple child nodes. Each child initially copies the parent's connections and inherits its feature and scale encoding, before a subsequent refinement step, then: *i*) redistributes the scale encoding among the children, *ii*) updates their features, and *iii*) removes edges that should not persist at the finer scale.

**Definition 2** (Bipartite graph expansion). Given a bipartite graph $B = (\mathcal{V}_L, \mathcal{V}_R, \mathcal{E}, \mathbf{b}, \mathbf{F}_L, \mathbf{F}_R)$ and expansion size vectors $\mathbf{v}_L \in \mathbb{N}^{|\mathcal{V}_L|}$, $\mathbf{v}_R \in \mathbb{N}^{|\mathcal{V}_R|}$, denoting the number of duplication for nodes (left side) and hyperedges (right side), respectively. Let $\tilde{B}(B, \mathbf{v}_L, \mathbf{v}_R) = (\tilde{\mathcal{V}}_L, \tilde{\mathcal{V}}_R, \tilde{\mathcal{E}}, \mathbf{b}^{\mathrm{expanded}}, \mathbf{F}_L^{\mathrm{expanded}}, \mathbf{F}_R^{\mathrm{expanded}})$ denote the expansion of $B$, whose node sets, scale encodings, and features are given by:

$$
\tilde{\mathcal{V}}_L = \bigcup_{p=1}^{|\mathcal{V}_L|} \mathcal{V}_L^{(p)}, \mathcal{V}_L^{(p)} = \{\, v_L^{(p,i)} \,\}_{i=1}^{\mathbf{v}_L[p]} \text{ for } 1 \le p \le |\mathcal{V}_L|,
$$

$$
\tilde{\mathcal{V}}_R = \bigcup_{p=1}^{|\mathcal{V}_R|} \mathcal{V}_R^{(p)}, \mathcal{V}_R^{(p)} = \{\, v_R^{(p,i)} \,\}_{i=1}^{\mathbf{v}_R[p]} \text{ for } 1 \le p \le |\mathcal{V}_R|,
$$

$$
\mathbf{b}^{\mathrm{expanded}}[p,i] = \mathbf{b}[p] \text{ for } 1 \le i \le \mathbf{v}_L[p], 1 \le p \le |\mathcal{V}_L|,
$$

$$
\mathbf{F}_L^{\mathrm{expanded}}[p,i] = \mathbf{F}_L[p] \text{ for } 1 \le i \le \mathbf{v}_L[p], 1 \le p \le |\mathcal{V}_L|,
$$

$$
\mathbf{F}_R^{\mathrm{expanded}}[p,i] = \mathbf{F}_R[p] \text{ for } 1 \le i \le \mathbf{v}_R[p], 1 \le p \le |\mathcal{V}_R|.
$$
(4)

The edge set $\tilde{\mathcal{E}}$ includes all the cluster interconnecting edges: $\{e_{\{p,i;q,j\}} \mid e_{\{p,q\}} \in \mathcal{E}, v_L^{(p,i)} \in \mathcal{V}_L^{(p)}, v_R^{(q,j)} \in \mathcal{V}_R^{(q)}\}$.

*Remark* 2. Expansion thus acts as a *clone-and-rewire* operation: vertices and hyperedges are duplicated, and each child inherits every incident edge of its parent.

**Definition 3** (Bipartite graph refinement). Given a bipartite graph $\tilde{B} = (\tilde{\mathcal{V}}_L, \tilde{\mathcal{V}}_R, \tilde{\mathcal{E}}, \mathbf{b}, \mathbf{F}_L^{\mathrm{expanded}}, \mathbf{F}_R^{\mathrm{expanded}})$, an edge selection vector $\mathbf{e} \in \{0,1\}^{|\tilde{\mathcal{E}}|}$, a scale encoding split vector $\mathbf{f} \in [0,1]^{|\tilde{\mathcal{V}}_L|}$, satisfying $\sum_{v \in \mathcal{V}_L^{(p)}} \mathbf{f}[v] = 1$ for all cluster $\mathcal{V}_L^{(p)}$ in $\tilde{\mathcal{V}}_L$, and two feature refinement vectors $\mathbf{F}_L^{\mathrm{refine}}$ and $\mathbf{F}_R^{\mathrm{refine}}$ with the same dimensions as $\mathbf{F}_L^{\mathrm{expanded}}$ and $\mathbf{F}_R^{\mathrm{expanded}}$, let $B(\tilde{B}, \mathbf{e}, \mathbf{f}, \mathbf{F}_L^{\mathrm{refine}}, \mathbf{F}_R^{\mathrm{refine}}) = (\tilde{\mathcal{V}}_L, \tilde{\mathcal{V}}_R, \mathcal{E}, \lceil \mathbf{b} \odot \mathbf{f} \rfloor, \mathbf{F}_L^{\mathrm{refine}}, \mathbf{F}_R^{\mathrm{refine}})$ denote the refinement of $\tilde{B}$, where $\mathcal{E} \subseteq \tilde{\mathcal{E}}$ such that the $i$-th edge $e_{(i)} \in \mathcal{E}$ if and only if $\mathbf{e}[i] = 1$.

*Remark* 3. Edges are selectively removed based on the binary indicator vector $\mathbf{e}$, and features are updated with new predictions. Node scale encodings are divided among child nodes according to the split proportions specified by the vector $\mathbf{f}$. Since scale encodings must be integers, the resulting values are rounded. In the case of a tie (*e.g.*, when an odd number must be split evenly), the child with the lowest index receives the largest share, and the remaining scale encoding is distributed among the others accordingly.

## 3.4. Probabilistic Modeling

We now present a formalization of our learning framework, generalizing (Bergmeister et al., 2024; Gailhard et al., 2025). Let $\{H^{(1)}, \ldots, H^{(N)}\}$ denote a set of *i.i.d.* hypergraph instances. Our objective is to approximate the unknown generative process by learning a distribution $p(H)$. We model the marginal likelihood of each hypergraph $H$ as a sum over the likelihoods of its bipartite representation's expansion sequences $p(H) = p(B) = \sum_{\varpi \in \Pi(B)} p(\varpi)$, where $\Pi(B)$ denotes the set of valid expansion sequences from a minimal bipartite graph to the full bipartite representation $B$ corresponding to $H$. Each intermediate $B^{(l-1)}$ is generated by expanding and refining its predecessor, in accordance with Definitions 2 and 3.

To simplify notations, we drop the superscript for $\mathbf{F}^{\mathrm{refine}}$ and simply write $\mathbf{F}$. Assuming a Markovian generative structure, and after various assumptions, we model the distribution of expansion and refinement sequences as:

$$
\begin{aligned}
p(\mathbf{v}_L^{(l)}, \mathbf{v}_R^{(l)}|B^{(l)})p(\mathbf{e}^{(l)}, \mathbf{f}^{(l)}, \mathbf{F}_L^{(l)}, \mathbf{F}_R^{(l)}|\tilde{B}^{(l)}) = \\
p(\mathbf{v}_L^{(l)}, \mathbf{v}_R^{(l)}, \mathbf{e}^{(l)}, \mathbf{f}^{(l)}, \mathbf{F}_L^{(l)}, \mathbf{F}_R^{(l)}|\tilde{B}^{(l)}),
\end{aligned}
$$
(5)

which corresponds to the combined expansion and refinement step that inverts coarsening at depth $l$. The detailed rationale for this modeling choice can be found in Appendix A.

During training, the model approximates the right-hand side of (5), learning to generate, for each expansion and refinement step, $\mathbf{v}_L$, $\mathbf{v}_R$, $\mathbf{e}$, $\mathbf{f}$, and $\mathbf{F}_L^{\mathrm{refine}}, \mathbf{F}_R^{\mathrm{refine}}$. Once trained, the model generates a hypergraph with $N$ nodes through successive expansion and refinement steps.

1. *Initialization.* Start from a minimal bipartite graph: $B^{(L)} = (\{1\}, \{2\}, \{(1,2)\}, (N))$, consisting of a single node on each side connected by one edge. The left-side node is assigned the full scale encoding, *i.e.*, $\mathbf{b} = [N]$. If node and hyperedge features need to be generated, $\mathbf{F}_L$ and $\mathbf{F}_R$ are initialized as zero matrices.

2. *Expansion and refinement.* Iteratively expand and refine the bipartite representation to add details until the target size is reached: $B^{(l)} \xrightarrow{\text{expand}} \tilde{B}^{(l-1)} \xrightarrow{\text{refine}} B^{(l-1)}$.

3. *Hypergraph reconstruction.* Once the final bipartite graph is generated, construct the hypergraph by collapsing each right-side node into a hyperedge connecting its adjacent left-side nodes.

Full details of the sampling procedure are provided in Algorithm 5 in Appendix G.

## 3.5. Multi-Scale Graph OT Coupling

Graphs and hypergraphs lack a canonical node ordering, making it challenging to align predictions with their targets. After expansion, the model may generate a permuted version of the correct output, leading to large losses despite correct predictions and introducing noise into the learning signal. To address this, we generalize *minibatch OT-coupling* (Tong et al., 2024; Pooladian et al., 2023) to align predictions and targets via a permutation that minimizes their matching cost. Unlike image generation, alignment in our setting must preserve graph structure and feature consistency.

Formally, for a given bipartite graph with $n$ nodes, adjacency matrix $\mathbf{A} \in \{0,1\}^{n \times n}$, expanded scale encodings $\mathbf{b} \in \mathbb{N}^n$ and expanded features $\mathbf{F} \in \mathbb{R}^{n \times d}$ where $d$ is the dimension of node or hyperedge features, $\mathbf{X} \in \mathbb{R}^{B \times d}$ and $\mathbf{Y} \in \mathbb{R}^{B \times d}$ correspond to prior samples and targets, respectively, the multi-scale graph OT coupling leads to the following optimization problem:

$$
\begin{aligned}
\mathbf{P}^* = \underset{\mathbf{P} \in \Pi_n}{\arg\min} \; &\|\mathbf{P}\mathbf{X} - \mathbf{Y}\|^2 \\
\text{s.t.} \quad &\mathbf{P}^\top \mathbf{A} \mathbf{P} = \mathbf{A}, \; \mathbf{P}\mathbf{b} = \mathbf{b}, \; \mathbf{P}\mathbf{F} = \mathbf{F},
\end{aligned}
\tag{6}
$$

where $\mathbf{P}$ is a permutation matrix and $\Pi_B$ denotes the set of all $B \times B$ permutations. To reduce overhead, we restrict permutations to children within a single cluster expansion, where equivalence naturally holds. With two or three children per cluster, only two or six permutations are possible, making the operation lightweight and easily parallelizable with standard tensor operations. We optimize (6) using Algorithm 2 in Appendix F.3.

**Proposition 2.** *Under the multi-scale graph OT coupling, the joint distribution $q(\mathbf{x}_0, \mathbf{x}_1)$ has marginals $q_0(\mathbf{x}_0), q_1(\mathbf{x}_1)$.*

Proposition 2 ensures that the multi-scale graph OT-coupling does not introduce bias in the learned distribution.

**Proposition 3.** *Let $\mathcal{B} = \{(\mathbf{x}_i^0, \mathbf{x}_i^1)\}_{i=1}^N$ be a minibatch of size $N$ coupled by a permutation $\pi$. Let $\Delta_\pi$ be the random variable representing the displacement vector $\mathbf{x}_i^1 - \mathbf{x}_{\pi(i)}^0$ for an index $i$ drawn uniformly from the batch. Multi-scale graph OT coupling (Algorithm 2) yields a permutation $\pi^*$ that reduces the total variance of this displacement:*

$$
\mathrm{tr}(\mathrm{Var}(\Delta_{\pi^*})) \leq \mathrm{tr}(\mathrm{Var}(\Delta_{Id})).
$$

Proposition 3 shows that the targets given by multi-scale graph OT coupling are more stable than uncoupled targets, leading to a more efficient training.

In practice, restricting permutations to local cluster expansions keeps the additional computational overhead of multi-scale graph OT coupling negligible (approximately $4\%$).

## 3.6. Generalization to Graphs

Our framework can be directly extended to generate standard graphs by adapting the coarsen–expand procedure in (Bergmeister et al., 2024). As in the hypergraph setting, each node begins with a scale encoding of one, which is summed when nodes are merged, while features are aggregated through a weighted average with weights proportional to the scale encodings. During expansion, clusters propagate their scale encodings and features to child nodes, and the model predicts how to split the scale encoding among them while refining their features. Graph generation, therefore, starts from a single node with a scale encoding equal to the desired size and proceeds through successive expansion and refinement steps. To ensure stable training and alignment between predicted and target structures, multi-scale graph OT coupling is applied to groups of expanded nodes exactly as in the hypergraph case.

## 4. Implementation

Our implementation relies on a conditional flow-matching framework (Lipman et al., 2023) combined with a *local PPGN* backbone (Bergmeister et al., 2024). At each refinement scale, the model jointly predicts: (*i*) node expansion decisions, (*ii*) hyperedge expansion decisions, (*iii*) incidence refinement variables, (*iv*) scale-encoding split proportions, and (*v*) node and hyperedge feature refinements.

Following an endpoint flow matching framework, the model learns a vector field transporting random initial states toward the target refinement variables. Expansion and incidence variables are initialized from Gaussian priors and mapped to continuous values in $[-1, 1]$, where the sign determines the corresponding discrete decision after thresholding. Scale-encoding split proportions are constrained to the simplex and initialized from Dirichlet distributions, while node and hyperedge features are initialized either from Gaussian or Dirichlet priors depending on the feature domain.

During training, each component is learned independently through a mean squared error objective between the predicted and target endpoints. During sampling, the learned flow is numerically integrated from $t = 0$ to $t = 1$ using Heun's second-order method.

To improve stability and concentrate model capacity on active generative regions, we leverage the hierarchical scale encoding to perform graph inpainting. Nodes and hyperedges that are not expanded inherit their parent connectivity and features deterministically, allowing the model to focus only on regions undergoing structural refinement. Concretely, during training, stationary nodes are masked out from the loss in order to avoid diluting the gradient with trivial identity mappings. At inference time, we similarly enforce several deterministic constraints: (*i*) clusters with

*Table 1.* Evaluation for hypergraph datasets: SBM, Ego, Tree, ModelNet40 Bookshelf, and ModelNet40 Piano.

| Method | SBM Hypergraphs ($n_{avg} = 31.73, std = 0.55$) | | Ego Hypergraphs ($n_{avg} = 109.71, std = 10.23$) | | Tree Hypergraphs ($n_{avg} = 32, std = 0$) | | ModelNet40 Bookshelf ($n_{avg} = 119.38, std = 68.20$) | | ModelNet40 Piano ($n_{avg} = 177.29, std = 57.11$) | |
|---|---|---|---|---|---|---|---|---|---|---|
| | Valid (%) ↑ | Spectral ↓ | Valid (%) ↑ | Spectral ↓ | Valid (%) ↑ | Spectral ↓ | Node Num ↓ | Spectral ↓ | Node Num ↓ | Spectral ↓ |
| HyperPA | 2.5 | 0.273 | 0.0 | 0.237 | 0.0 | 0.159 | 8.025 | 0.048 | 0.825 | 0.067 |
| VAE | 0.0 | 0.024 | 0.0 | 0.133 | 0.0 | 0.124 | 47.450 | 0.190 | 75.350 | 0.396 |
| GAN | 0.0 | 0.059 | 0.0 | 0.230 | 0.0 | 0.089 | **0.000** | 0.476 | **0.000** | 0.697 |
| Diffusion | 0.0 | 0.031 | 0.0 | 0.190 | 0.0 | 0.127 | **0.000** | 0.079 | 0.050 | 0.113 |
| HYGENE | 65.0 | 0.010 | 90.0 | **0.004** | 77.5 | 0.012 | 69.730 | 0.068 | 42.520 | 0.117 |
| FAHNES | **87.8±3.1** | **0.006±0.004** | **99.5±1.1** | 0.004±0.003 | **89.7±6.0** | **0.003±0.002** | 0.135±0.276 | **0.024±0.015** | 0.846±1.009 | **0.040±0.026** |

*Table 2.* Evaluation for graph datasets: SBM, Tree, Planar, Protein, and Point Cloud.

| Method | SBM graphs ($n_{avg} = 105.99, std = 38.38$) | | Tree graphs ($n_{avg} = 64.0, std = 0$) | | Planar graphs ($n_{avg} = 64.0, std = 0$) | | Protein ($n_{avg} = 261.5, std = 104.7$) | | Point cloud ($n_{avg} = 1434.3, std = 1285.9$) | |
|---|---|---|---|---|---|---|---|---|---|---|
| | Valid (%) ↑ | Ratio ↓ | Valid (%) ↑ | Ratio ↓ | Valid (%) ↑ | Ratio ↓ | Spectral ↓ | Ratio ↓ | Spectral ↓ | Ratio ↓ |
| HSpectre | 45.0 | 10.2 | **100.0** | 4.0 | 95.0 | 2.1 | **0.001** | 5.9 | **0.005** | 7.0 |
| BwR | 7.5 | 38.6 | 0.0 | 11.4 | 0.0 | 251.9 | 0.070 | 254.4 | 0.291 | 133.2 |
| DiGress | 60.0 | **1.7** | 90.0 | **1.6** | 77.5 | 5.1 | 0.002 | 18.0 | OOM | OOM |
| DeFoG | **90.0±5.1** | 4.9±1.3 | 96.5±2.6 | 1.6±0.4 | **99.5±1.0** | **1.6±0.04** | — | — | — | — |
| FAHNES | 50.0±5.0 | 4.8±0.7 | **100.0±0.0** | 1.8±0.8 | 96.7±6.2 | 2.2±1.6 | **0.001±0.001** | 3.8±1.9 | **0.005±0.001** | **3.2±0.5** |

scale encoding 1 cannot expand; (*ii*) clusters of size 2 are forced to split evenly; (*iii*) non-expanded nodes preserve their scale encodings; and (*iv*) features of non-expanded nodes and hyperedges are propagated unchanged.

To ensure consistency of the generated scale encodings, predicted split proportions are projected onto the simplex within each expanded cluster. Likewise, when simplex-valued node or hyperedge features are used, they are projected back onto the simplex after each refinement step.

Graph inpainting is motivated by the following result:

**Proposition 4.** *Let* $\mathbf{x}^{(l)}$ *be the upsampled node state at scale* $l$ *and* $\mathbf{y}^{(l)}$ *be the ground-truth target. In hierarchical generation, the node set* $\mathcal{V}$ *is partitioned into static nodes* $\mathcal{S} = \{v : \mathbf{y}_v^{(l)} = \mathbf{x}_v^{(l)}\}$ *and active nodes* $\mathcal{A} = \{v : \mathbf{y}_v^{(l)} \neq \mathbf{x}_v^{(l)}\}$. *As hierarchy depth increases, the gradient signal* $\nabla_\theta \mathcal{L}$ *becomes concentrated around learning a trivial identity mapping due to the asymptotic dominance of* $\mathcal{S}$ ($|\mathcal{S}| \gg |\mathcal{A}|$). *Restricting the loss support to the active nodes* $\mathcal{A}$ *recovers the generative signal.*

Intuitively, as refinement progresses, most nodes already match their target states, creating a strong bias toward learning a trivial identity mapping. The comparatively sparse active refinements therefore become diluted in the global objective. Masking stationary regions removes this dominant background signal and focuses the model on the local structural updates at the refinement frontier.

Finally, we apply the proposed multi-scale graph OT coupling before interpolation in the flow-matching process to align duplicated nodes and hyperedges within each expanded cluster while preserving structural consistency.

The hierarchical strategy ensures high efficiency; the total computational complexity of our approach scales as $\mathcal{O}((n + m + k)\log n)$ for a hypergraph with $n$ nodes, $m$ hyperedges, and $k$ incidences. Additional implementation details regarding training, sampling, self-conditioning, and complexity are provided in Appendices F, G, and H.

## 5. Experiments and Results

In this section, we detail our experimental setup, covering datasets, evaluation metrics, baselines, results, and ablation studies. Full experimental details are provided in Appendix E. Visualizations of the generated samples are provided in Appendix J.

### 5.1. Datasets and Experimental Setup

**Datasets**. For featureless structures, we evaluate our method on five hypergraph datasets: Stochastic Block Model (*SBM*) (Kim et al., 2018), *Ego* (Comrie & Kleinberg, 2021), *Tree* (Nieminen & Peltola, 1999), and ModelNet40 *bookshelf* and *piano* (Wu et al., 2015). We also evaluate FAHNES on five unfeatured graph datasets from (Bergmeister et al., 2024): *SBM*, *Tree*, *Planar*, *Protein*, and *Point cloud*. Regarding featured structures, we evaluate FAHNES on two featured hypergraph datasets consisting of 3D meshes: Manifold40 *bench* and *airplane* (Hu et al., 2022); and two graph point cloud datasets sampled from the same mesh categories. For 3D mesh and point cloud datasets, node features are 3D positions, while edges and hyperedges do not have features.

**Metrics**. For featureless hypergraphs, we follow the evaluation protocol of (Gailhard et al., 2025), including: *i*) struc-

*Table 3.* Evaluation on the graph point cloud datasets.

| Method | Airplane Point Clouds ($n_{avg} = 971.14$, $std = 63.15$) | | | Bench Point Clouds ($n_{avg} = 987.31$, $std = 49.33$) | | |
|---|---|---|---|---|---|---|
| | ChamDist ↓ | ChamCov (%) ↑ | ChamDiv ↑ | ChamDist ↓ | ChamCov (%) ↑ | ChamDiv ↑ |
| Flow Matching | OOM | OOM | OOM | OOM | OOM | OOM |
| DiGress | OOM | OOM | OOM | OOM | OOM | OOM |
| DeFoG | OOM | OOM | OOM | OOM | OOM | OOM |
| FAHNES | **0.094**±0.006 | **22.2**±14.5 | **0.085**±0.026 | **0.130**±0.001 | **15.8**±10.6 | **0.191**±0.114 |

*Table 4.* Evaluation on the hypergraph ManifoldNet meshes.

| Method | ManifoldNet Airplane (Mesh) ($n_{avg} = 51.86$, $std = 0.74$) | | | ManifoldNet Bench (Mesh) ($n_{avg} = 48.90$, $std = 3.52$) | | |
|---|---|---|---|---|---|---|
| | ChamDist ↓ | ChamCov (%) ↑ | ChamDiv ↑ | ChamDist ↓ | ChamCov (%) ↑ | ChamDiv ↑ |
| Sequential | 0.082±0.074 | **34.8**±0.0 | 0.129±0.011 | 0.169±0.007 | 13.3±0.0 | 0.157±0.003 |
| WGAN | 0.124±0.021 | 10.4±3.5 | 0.037±0.016 | 0.128±0.015 | 17.3±3.3 | 0.063±0.099 |
| Flow Matching | 0.085±0.014 | 20.9±10.0 | 0.063±0.005 | 0.108±0.019 | 30.7±2.5 | 0.112±0.007 |
| VAE | 0.231±0.208 | 26.9±11.7 | **0.191**±0.130 | 0.280±0.223 | 27.5±9.9 | 0.195±0.107 |
| FAHNES | **0.048**±0.003 | 31.9±13.2 | 0.101±0.011 | **0.064**±0.005 | **42.2**±13.9 | **0.243**±0.037 |

*Table 5.* Ablation studies on the scale encodings (Enc.) and multi-scale graph OT-coupling (Coup.) for ModelNet and ManifoldNet.

| Enc. | Coup. | ModelNet Bookshelf | | ModelNet Piano | | ManifoldNet Airplane | | | ManifoldNet Bench | | |
|---|---|---|---|---|---|---|---|---|---|---|---|
| | | Node Num ↓ | Spectral ↓ | Node Num ↓ | Spectral ↓ | ChamDist ↓ | ChamCov (%) ↑ | ChamDiv ↑ | ChamDist ↓ | ChamCov (%) ↑ | ChamDiv ↑ |
| ✓ | ✓ | **0.135**±0.276 | 0.024±0.015 | **0.846**±1.009 | 0.040±0.026 | **0.048**±0.003 | 31.9±13.2 | 0.101±0.011 | **0.064**±0.005 | 42.2±13.9 | **0.243**±0.037 |
| ✗ | ✓ | 0.940±0.917 | 0.032±0.013 | 3.622±1.822 | 0.055±0.040 | 0.079±0.019 | 29.0±17.6 | 0.080±0.011 | 0.090±0.003 | 37.8±16.8 | 0.232±0.048 |
| ✓ | ✗ | 0.265±0.496 | **0.014**±0.007 | 3.155±3.637 | **0.030**±0.048 | 0.050±0.005 | 30.4±15.0 | 0.093±0.019 | 0.085±0.056 | 32.2±7.7 | 0.208±0.030 |
| ✗ | ✗ | 1.325±1.631 | 0.031±0.009 | 5.490±8.847 | 0.036±0.016 | 0.100±0.023 | **44.9**±10.0 | **0.114**±0.031 | 0.098±0.024 | **42.2**±22.0 | 0.235±0.053 |

tural metrics such as *Node Num* (node count difference), and *ii*) topological metrics such as *Spectral* computed as the maximum mean discrepancy (MMD) between spectral distributions. When datasets impose structural constraints, we also report *Valid*, the percentage of valid generated samples. Lower values indicate better performance for all metrics except *Valid*, where higher is better. For 3D meshes and point clouds, we report *ChamDist*, the minimum Chamfer distance to training samples, *ChamCov*, the fraction of reference samples matched by at least one generated sample, and *ChamDiv*, the average pairwise Chamfer distance between generated samples. Lower values of *ChamDist* indicate better performance, while higher values of *ChamCov* and *ChamDiv* indicate better coverage and diversity, respectively. For graph datasets, we also report the ratio of several metrics between generated and test sets (see Appendix E for details). Additional metrics and full numerical results are provided in Appendix I.

**Baselines.** For the featureless hypergraphs, we compare FAHNES against HYGENE (Gailhard et al., 2025), HyperPA (Do et al., 2020), a Variational Autoencoder (VAE) (Kingma & Welling, 2014), a Generative Adversarial Network (GAN) (Goodfellow et al., 2020), and a standard 2D diffusion model (Ho et al., 2020) trained on incidence matrix images, where hyperedge membership is represented by white pixels and absence by black pixels. For 3D meshes, we compare FAHNES with Wasserstein GAN (WGAN) (Arjovsky et al., 2017), flow matching, and VAE trained in the concatenation of the incidence matrix plus features, along with a sequential disjoint generation baseline where the hypergraph topology is generated first, followed by the feature generation. For the graph datasets, we compare FAHNES against DeFoG (Qin et al., 2025) and DiGress (Vignac et al., 2023), two state-of-the-art flat models for graph generation, BwR (Diamant et al., 2023), a recent graph bandwidth restriction method intended for scalability, and HSpectre (Bergmeister et al., 2024), a hierarchical gen-

eration method for unfeatured graphs. All baseline results are taken from (Qin et al., 2025; Bergmeister et al., 2024).

## 5.2. Results and Discussion

**Comparison with the baselines.** Table 1 shows the comparisons for the featureless hypergraphs, where **bold** and underlined indicate the best and second-best scores. Our scale encodings and multi-scale graph OT coupling consistently improve generation quality. About the featureless graphs, Table 2 shows that FAHNES obtains competitive results compared to state-of-the-art flat methods on small-graph datasets, and that our new components improve generation quality in all datasets compared to HSpectre. On featured point clouds (Table 3), our hierarchical approach is the only method that scales, highlighting its quasi-linear complexity compared to the quadratic cost of flat models. Regarding the mesh-featured hypergraph datasets, Table 4 shows that FAHNES obtains better results in general than other baselines like the sequential generation or naive-joint approaches through concatenation (WGAN, Flow Matching, and VAE). This shows the necessity of jointly generating topology and features in hypergraphs instead of simple two-step or naive-joint approaches. Detailed numerical results are shown in Appendix I.

## 5.3. Ablation Studies

Table 5 presents ablation studies of FAHNES for the scale encodings and multi-scale graph OT coupling components. We observe that using scale encodings instead of concatenating the target size to each node embedding, like in (Bergmeister et al., 2024; Gailhard et al., 2025), improves generation quality. The multi-scale graph OT coupling has a more nuanced effect, clearly improving quality on some datasets, like ManifoldNet Airplane, while not changing much for others. Those two components are also essential for feature generation, as lacking one of them results

in much worse results, as seen by the large increase in *ChamDist* when one component is ablated. We present an additional ablation regarding the coarsening in Appendix I.

## 5.4. Limitations

While our scale encoding helps mitigate the issue of missing nodes, it does not fully resolve it, and our method still struggles when generating very large hypergraphs, such as ModelNet Bookshelf and Piano, or very large graphs, such as point cloud datasets. These cases highlight that scalability remains challenging when both the number of nodes and the structural complexity grow substantially. Moreover, our framework currently assumes continuous feature distributions (*e.g.*, 3D coordinates), and extending it to domains with richer or heterogeneous node and hyperedge attributes (such as categorical or multi-modal features) may require additional modeling components. Finally, although FAHNES improves stability through multi-scale graph OT coupling, training remains computationally expensive for large-scale datasets, which may limit applicability in resource-constrained settings.

## 6. Conclusion

We presented FAHNES, which is, to the best of our knowledge, the first scalable hierarchical framework able to generate not only topology but also features, for both hypergraphs and graphs. By integrating coarse-to-fine structural modeling with feature-aware generation, FAHNES overcomes the limitations of flat or disjoint approaches and enables the generation of complex structures at larger scales. Our main innovations include a scale encoding, which provides fine-grained control over local growth and improves the training signal of the gradient, and multi-scale graph OT coupling, which improves alignment and stability during training. Extensive experiments on synthetic, protein, 3D mesh, and point cloud datasets show that FAHNES achieves strong performance in both topology and feature generation while scaling to graph and hypergraph sizes that are out of reach for prior models. Our framework opens new directions for generative modeling of structured data, including applications in 3D geometry and circuit modeling, as well as extensions to richer feature modalities and more scalable generative models for structured data.

## Impact Statement

This paper presents work whose goal is to advance the field of machine learning. There are many potential societal consequences of our work, none of which we feel must be specifically highlighted here.

## Acknowledgments

The authors acknowledge the French National Research Agency (ANR) for its financial support of the System On Chip Design leveraging Artificial Intelligence (SODA) project under grant ANR-23-IAS3-0004 and the JCJC project DeSNAP ANR-24-CE23-1895-01. This work is also supported by Hi! PARIS and ANR/France 2030 program ANR-23-IACL-0005.

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

The appendices offer additional technical details and formal proofs to complement the main paper. The document is structured as follows: Our detailed rationale for the probabilistic modeling is presented in Appendix A. Formal proofs of the main propositions are presented in Appendix B. Appendix C describes our procedure for sampling coarsening sequences. Appendix D provides illustrative examples of coarsening sequences. Appendix E outlines the experimental setup, including hyperparameter choices and numerical results. Appendix F discusses more detailed implementation details of FAHNES. Algorithms for model training and sampling featured hypergraphs are detailed in Appendix G. Appendix H analyzes the algorithmic complexity of our approach. Appendix I presents detailed numerical results for the ablation studies. Appendix J presents visual comparisons between training and generated samples.

## A. Probabilistic Modeling

In this section, we present a formalization of our learning framework, generalizing (Bergmeister et al., 2024; Gailhard et al., 2025). Let $\{H^{(1)}, \ldots, H^{(N)}\}$ denote a set of *i.i.d.* hypergraph instances. Our objective is to approximate the unknown generative process by learning a distribution $p(H)$. We model the marginal likelihood of each hypergraph $H$ as a sum over the likelihoods of its bipartite representation's expansion sequences $p(H) = p(B) = \sum_{\varpi \in \Pi(B)} p(\varpi)$, where $\Pi(B)$ denotes the set of valid expansion sequences from a minimal bipartite graph to the full bipartite representation $B$ corresponding to $H$. Each intermediate $B^{(l-1)}$ is generated by expanding and refining its predecessor, in accordance with Definitions 2 and 3.

To simplify notations, we drop the superscript for $\mathbf{F}^{\text{refine}}$ and simply write $\mathbf{F}$. Assuming a Markovian generative structure, the likelihood of a specific expansion sequence $\varpi$ is factorized as:

$$p(\varpi) = \overbrace{p(B^{(L)})}^{1} \prod_{l=L}^{1} p(B^{(l-1)}|B^{(l)}) = \prod_{l=L}^{1} p(\mathbf{e}^{(l-1)}, \mathbf{f}^{(l-1)}, \mathbf{F}_L^{(l-1)}, \mathbf{F}_R^{(l-1)}|\tilde{B}^{(l-1)}) p(\mathbf{v}_L^{(l)}, \mathbf{v}_R^{(l)}|B^{(l)}). \qquad (7)$$

To simplify the modeling process and avoid learning two separate distributions $p(\mathbf{e}^{(l)}, \mathbf{f}^{(l)}, \mathbf{F}_L^{(l)}, \mathbf{F}_R^{(l)}|\tilde{B}^{(l)})$ and $p(\mathbf{v}_L^{(l)}, \mathbf{v}_R^{(l)}|B^{(l)})$, we rearrange terms as follows:

$$p(\varpi) = p(\mathbf{e}^{(0)}, \mathbf{f}^{(0)}, \mathbf{F}_L^{(0)}, \mathbf{F}_R^{(0)}|\tilde{B}^0) p(\mathbf{v}_L^{(L)}, \mathbf{v}_R^{(L)}) \left[ \prod_{l=L-1}^{1} p(\mathbf{v}_L^{(l)}, \mathbf{v}_R^{(l)}|B^{(l)}) p(\mathbf{e}^{(l)}, \mathbf{f}^{(l)}, \mathbf{F}_L^{(l)}, \mathbf{F}_R^{(l)}|\tilde{B}^{(l)}) \right], \qquad (8)$$

where $p(\mathbf{v}_L^{(L)}, \mathbf{v}_R^{(L)}) = p(\mathbf{v}_L^{(L)}, \mathbf{v}_R^{(L)}|B^{(L)})$.

We assume that the variables $\mathbf{v}_L^{(l)}$ and $\mathbf{v}_R^{(l)}$ are conditionally independent of $\tilde{B}^{(l)}$ when conditioned on $B^{(l)}$:

$$p(\mathbf{v}_L^{(l)}, \mathbf{v}_R^{(l)}|B^{(l)}, \tilde{B}^{(l)}) = p(\mathbf{v}_L^{(l)}, \mathbf{v}_R^{(l)}|B^{(l)}). \qquad (9)$$

This allows us to write the combined likelihood as:

$$p(\mathbf{v}_L^{(l)}, \mathbf{v}_R^{(l)}|B^{(l)}) p(\mathbf{e}^{(l)}, \mathbf{f}^{(l)}, \mathbf{F}_L^{(l)}, \mathbf{F}_R^{(l)}|\tilde{B}^{(l)}) = p(\mathbf{v}_L^{(l)}, \mathbf{v}_R^{(l)}, \mathbf{e}^{(l)}, \mathbf{f}^{(l)}, \mathbf{F}_L^{(l)}, \mathbf{F}_R^{(l)}|\tilde{B}^{(l)}), \qquad (10)$$

which corresponds to the combined expansion and refinement step that inverts coarsening at depth $l$.

## B. Proofs

### B.1. Optimal Downsampled Features

**Proposition.** *Let $\mathbf{X}$ be the node feature matrix and $\mathcal{P} = \{\mathcal{C}_k\}_{k=1}^{K}$ a partition of the vertices into clusters. The optimal downsampled features minimizing the reconstruction error is the cluster-wise barycenter:*

$$\mathbf{x}_{\mathcal{C}_k}^* = \frac{1}{|\mathcal{C}_k|} \sum_{v \in \mathcal{C}_k} \mathbf{x}_v \quad \textit{for } 1 \le k \le K.$$

*Proof.* Let $H = (\mathcal{V}, \mathcal{E})$ be a hypergraph with node set $\mathcal{V}$ and hyperedge set $\mathcal{E}$, and let $H_l = (\mathcal{V}_l, \mathcal{E}_l)$ denote the lifted hypergraph obtained by expanding each cluster $\mathcal{C} \subseteq \mathcal{V}$ into a $|C|$-clique. By construction, there exists a bijection $\phi : \mathcal{V} \to \mathcal{V}_l$

mapping each original node to its corresponding lifted node. Suppose each node $v \in \mathcal{V}$ is associated with a feature vector $\mathbf{x}_v \in \mathbb{R}^d$, and each cluster $\mathcal{C} \subseteq \mathcal{V}$ is assigned a cluster feature vector $\mathbf{x}_\mathcal{C} \in \mathbb{R}^d$, which is inherited by all lifted nodes $\phi(v)$ for $v \in \mathcal{C}$. Define the mean squared error between the original node features and the cluster features in the lifted hypergraph as:

$$\text{MSE} = \sum_{v \in \mathcal{V}} \|\mathbf{x}_v - \mathbf{x}_{C(v)}\|^2, \tag{11}$$

where $C(v)$ denotes the cluster containing node $v$.

To minimize the MSE, it suffices to minimize, for each cluster $\mathcal{C}$,

$$J_\mathcal{C}(\mathbf{x}_\mathcal{C}) = \sum_{v \in \mathcal{C}} \|\mathbf{x}_v - \mathbf{x}_\mathcal{C}\|^2. \tag{12}$$

Since $J_\mathcal{C}$ is a convex quadratic function in $\mathbf{x}_\mathcal{C}$, we find its minimum by setting the gradient to zero:

$$\nabla_{\mathbf{x}_\mathcal{C}} J_\mathcal{C}(\mathbf{x}_C) = \sum_{v \in \mathcal{C}} 2(\mathbf{x}_\mathcal{C} - \mathbf{x}_v) = 2|\mathcal{C}|\mathbf{x}_\mathcal{C} - 2\sum_{v \in \mathcal{C}} \mathbf{x}_v = \mathbf{0}. \tag{13}$$

Solving for $\mathbf{x}_\mathcal{C}$, we obtain

$$\mathbf{x}_\mathcal{C} = \frac{1}{|\mathcal{C}|} \sum_{v \in \mathcal{C}} \mathbf{x}_v, \tag{14}$$

which is the arithmetic mean of the feature vectors in the cluster $\mathcal{C}$. $\square$

## B.2. Multi-Scale Graph OT Coupling

Let $B$ be a bipartite graph. Let $\mathbf{b}$ be its current scale encoding, $\mathbf{F}_L$ its node features, and $\mathbf{F}_R$ its hyperedge features. Denote $\mathbf{x} = (\mathbf{v}_L, \mathbf{v}_R, \mathbf{e}, \mathbf{f}, \mathbf{F}_L^{\text{refine}}, \mathbf{F}_R^{\text{refine}})$, *i.e.*, the predictions of the model. We will denote $\mathbf{x}_0$ the initial noise samples and $\mathbf{x}_1$ the targets. In the following, all distributions are conditioned on $\mathbf{b}$, $\mathbf{F}_L$ and $\mathbf{F}_R$.

**Proposition.** *Under the OT-coupling of Algorithm 2, the joint distribution:*

$$q(\mathbf{x}_0, \mathbf{x}_1)$$

*has marginals:*

$$q_0(\mathbf{x}_0), \quad q_1(\mathbf{x}_1).$$

**Definition 4** (Isomorphism)**.** Let

$$B_1 = (\mathcal{V}_L^1, \mathcal{V}_R^1, \mathcal{E}^1, \mathbf{b}^1, \mathbf{F}_L^1, \mathbf{F}_R^1), \quad B_2 = (\mathcal{V}_L^2, \mathcal{V}_R^2, \mathcal{E}^2, \mathbf{b}^2, \mathbf{F}_L^2, \mathbf{F}_R^2),$$

be bipartite graphs.

We say $B_1 \cong B_2$ (are isomorphic) if there exist bijections

$$\sigma_L \colon \mathcal{V}_L^1 \to \mathcal{V}_L^2, \quad \sigma_R \colon \mathcal{V}_R^1 \to \mathcal{V}_R^2$$

such that:

1. Edge structure is preserved: $(v, w) \in \mathcal{E}^1 \iff (\sigma_L(v), \sigma_R(w)) \in \mathcal{E}^2$,
2. Scale encodings are preserved: $\mathbf{b}^1(v) = \mathbf{b}^2(\sigma_L(v))$ for all $v \in \mathcal{V}_L^1$, $\quad \mathbf{b}^1(w) = \mathbf{b}^2(\sigma_R(w))$ for all $w \in \mathcal{V}_R^1$,
3. Node and hyperedge features are preserved: $\mathbf{F}_L^1(v) = \mathbf{F}_L^2(\sigma_L(v)), \quad \mathbf{F}_R^1(w) = \mathbf{F}_R^2(\sigma_R(w))$.

*Proof.* Let $B = (\mathcal{V}_L, \mathcal{V}_R, \mathcal{E}, \mathbf{x}_1)$ be an arbitrary target bipartite graph, and $f$ be an arbitrary test function defined on bipartite graphs. To alleviate notations, we will denote $f(\mathbf{x})$ the value of $f$ for the same bipartite graph where values of $\mathbf{x}_1$ are replaced by those of $\mathbf{x}$.

Algorithm 2 swaps noise samples between nodes that are equivalent in the sense that the original and swapped graphs (conditioned on the same topology and features) are isomorphic. That is, if $\mathbf{x}_0$ and $\mathbf{x}_0'$ differ only by such a swap, then the

resulting bipartite graphs are isomorphic. Let $\sigma$ be the bijective reindexing function corresponding to this isomorphism. Since $f$ is defined on graphs and graphs are invariant under isomorphism, we have:

$$f(\mathbf{x}_0) = f(\sigma(\mathbf{x}_0)). \tag{15}$$

Therefore:

$$\mathbb{E}_{q(\mathbf{x}_0, \mathbf{x}_1)}[f(\mathbf{x}_0)] = \mathbb{E}_{q(\mathbf{x}_1)}\left[\mathbb{E}_{q(\mathbf{x}_0|\mathbf{x}_1)}[f(\mathbf{x}_0)]\right] \tag{16}$$

$$= \mathbb{E}_{q(\mathbf{x}_1)}\left[\mathbb{E}_{q(\mathbf{x}_0)}[f(\sigma(\mathbf{x}_0))]\right] \tag{17}$$

$$= \mathbb{E}_{q(\mathbf{x}_1)}\left[\mathbb{E}_{q(\mathbf{x}_0)}[f(\mathbf{x}_0)]\right] \tag{18}$$

$$= \mathbb{E}_{q(\mathbf{x}_0)}[f(\mathbf{x}_0)]. \tag{19}$$

where (17) comes from the transfer formula. Please note that in the above equations, $\sigma$ is specific to the couple $(\mathbf{x}_0, \mathbf{x}_1)$.

Thus, the marginal distribution over $\mathbf{x}_0$ remains unchanged. The same argument applies symmetrically for $\mathbf{x}_1$ by using $\sigma^{-1}$, concluding the proof. $\square$

**Proposition.** *Let $\mathcal{B} = \{(\mathbf{x}_i^0, \mathbf{x}_i^1)\}_{i=1}^N$ be a minibatch of size $N$ coupled by a permutation $\pi$. Let $\Delta_\pi$ be the random variable representing the displacement vector $\mathbf{x}_i^1 - \mathbf{x}_{\pi(i)}^0$ for an index $i$ drawn uniformly from the batch. Multi-scale graph OT-coupling (Algorithm 2) yields a permutation $\pi^*$ that reduces the total variance of this displacement:*

$$\mathrm{tr}(\mathrm{Var}(\Delta_{\pi^*})) \leq \mathrm{tr}(\mathrm{Var}(\Delta_{Id})).$$

*Proof.* The total variance decomposes into:

$$\mathrm{Tr}(\mathrm{Var}(\Delta_\pi)) = \mathbb{E}[\|\Delta_\pi\|^2] - \|\mathbb{E}[\Delta_\pi]\|^2. \tag{20}$$

First, by the linearity of expectation, we have:

$$\mathbb{E}[\Delta_\pi] = \mathbb{E}[\mathbf{x}^1] - \mathbb{E}[\mathbf{x}_\pi^0].$$

Since the permutation $\pi$ only reorders the samples within the minibatch without changing the set of values, the marginal expectations $\mathbb{E}[\mathbf{x}^1]$ and $\mathbb{E}[\mathbf{x}_\pi^0] = \mathbb{E}[\mathbf{x}^0]$ do not depend on the coupling. Thus, this term is constant.

Second, we can rewrite:

$$\mathbb{E}[\|\Delta_\pi\|^2] \propto \sum_{i=1}^N \|\mathbf{x}_i^1 - \mathbf{x}_{\pi(i)}^0\|^2 = \sum_{\text{clusters } \mathcal{C}} \sum_{i \in \mathcal{C}} \|\mathbf{x}_i^1 - \mathbf{x}_{\pi(i)}^0\|^2.$$

Algorithm 2 is constructed to minimize the $\sum_{i \in \mathcal{C}} \|\mathbf{x}_i^1 - \mathbf{x}_{\pi(i)}^0\|^2$ locally within each cluster. Consequently, we have:

$$\mathbb{E}[\|\Delta_{\pi^*}\|^2] \leq \mathbb{E}[\|\Delta_{Id}\|^2].$$

Since the first term is reduced and the second term is constant, the total variance $\mathrm{Tr}(\mathrm{Var}(\Delta_{\pi^*}))$ is reduced. $\square$

### B.3. Preservation of the Gradient Signal

**Proposition.** *Let $\mathbf{x}^{(l)}$ be the upsampled node state at scale $l$ and $\mathbf{y}^{(l)}$ be the ground-truth target. In hierarchical generation, the node set $\mathcal{V}$ is partitioned into static nodes $\mathcal{S} = \{v : \mathbf{y}_v^{(l)} = \mathbf{x}_v^{(l)}\}$ and active nodes $\mathcal{A} = \{v : \mathbf{y}_v^{(l)} \neq \mathbf{x}_v^{(l)}\}$. As hierarchy depth increases, the gradient signal $\nabla_\theta \mathcal{L}$ becomes concentrated around learning a trivial identity mapping due to the asymptotic dominance of $\mathcal{S}$ ($|\mathcal{S}| \gg |\mathcal{A}|$). Restricting the loss support to the active nodes $\mathcal{A}$ recovers the generative signal.*

*Proof.* Let $\mathbf{x}^{(l)}$ be the upsampled state serving as input for scale $l$, and $\mathbf{y}^{(l)}$ be the ground truth target. We partition the node set $\mathcal{V}$ into active nodes $\mathcal{A}$ (where refinement is required, $\mathbf{y}_v \neq \mathbf{x}_v$) and static nodes $\mathcal{S}$ (where the coarse approximation is exact, $\mathbf{y}_v = \mathbf{x}_v$).

A standard global objective minimizes the MSE over the entire graph:

$$\mathcal{L}_{\text{total}} = \overbrace{\sum_{v \in \mathcal{S}} \|f_\theta(\mathbf{x}_t, \mathbf{x}_v) - \mathbf{x}_v\|_2^2}^{\text{Identity Reconstruction}} + \overbrace{\sum_{v \in \mathcal{A}} \|f_\theta(\mathbf{x}_t, \mathbf{x}_v) - \mathbf{y}_v\|_2^2}^{\text{Structural Refinement}}, \tag{21}$$

where $\mathbf{x}_t$ is the denoised sample at time $t$. As generation deepens, $|\mathcal{S}| \to N$ while $|\mathcal{A}|$ becomes small. Consequently, the Identity Reconstruction term dominates the loss landscape. Minimizing this term encourages the model to learn the trivial solution $f_\theta(\mathbf{x}_t, \mathbf{x}) \approx \mathbf{x}$ globally. Thus, the gradients required to learn the complex refinement in $\mathcal{A}$ are overwhelmed by the gradients enforcing stationarity in $\mathcal{S}$.

By using the scale encoding to mask $\mathcal{S}$ (where $\mathbf{b} = 1$), the objective reduces to:

$$\mathcal{L}_{\text{masked}} = \sum_{v \in \mathcal{A}} \|f_\theta(\mathbf{x}_t, \mathbf{x}_v) - \mathbf{y}_v\|_2^2. \tag{22}$$

This removes the identity bias entirely, ensuring that the capacity of the model and gradient updates are driven solely by the non-trivial generative discrepancies in the active regions. $\qquad\square$

## C. Coarsening Sequence Sampling

This section outlines our methodology for sampling a coarsening sequence $\pi \in \Pi_F(H)$ for a given hypergraph $H$. The full procedure is detailed in Algorithm 1. At each coarsening step $l$, let $H^{(l)}$ denote the current hypergraph, $B^{(l)}$ its bipartite representation, and $C^{(l)}$ its weighted clique expansion. We begin by sampling a target reduction fraction red_frac $\sim \mathcal{U}([\rho_{\min}, \rho_{\max}])$. We then evaluate all possible contraction sets $F(C^{(l-1)})$ using a cost function $c$, where lower cost indicates higher preference. We employ a greedy randomized strategy that processes contraction sets in order of increasing cost. For each set:

- The set is stochastically rejected with probability $1 - \lambda$.
- If not rejected:
  - **Overlap check:** If the contraction set overlaps with any previously accepted contraction, it is discarded.
  - **Coarsening attempt:** Otherwise, we compute tentative coarsened representations $C_{\text{temp}}$ and $B_{\text{temp}}$.
  - **Cluster constraint check:** If all right-side clusters in $B_{\text{temp}}$ contain at most three nodes, the contraction is accepted.
  - **Update step:** When a contraction is accepted, we:
    * Sum the scale encodings of the nodes in the contraction set to define the new cluster scale encoding.
    * Compute the new cluster's node features as a weighted average of the original features, using node scale encodings as weights.

The loop terminates once the number of remaining nodes satisfies the stopping condition:

$$|\mathcal{V}_L^{(l-1)}| - |\bar{\mathcal{V}}_L^{(l)}| > \text{red\_frac} \cdot |\mathcal{V}_L^{(l-1)}|,$$

*i.e.*, when the number of nodes on the left side (corresponding to the original hypergraph nodes) has been reduced by the sampled fraction. This framework allows a variety of cost functions $c$, contraction families $F$, reduction fraction ranges $[\rho_{\min}, \rho_{\max}]$, and randomization parameters $\lambda$.

**Practical considerations.** To avoid oversampling overly small graphs during training, we follow the heuristic of (Bergmeister et al., 2024): when the current graph has fewer than 16 nodes, we fix the reduction fraction to $\rho = \rho_{\max}$. Due to the constraint that no right-side cluster in $B^{(l)}$ may contain more than three nodes, achieving the target reduction fraction is not always feasible. However, we observe empirically that this rarely poses a problem when $\rho_{\max}$ is reasonably small.

During training, we sample a coarsening sequence for each dataset graph, but only retain a randomly selected intermediate graph from the sequence. Thus, our practical implementation of Algorithm 1 is designed to return a single coarsened graph with associated features and scale encodings, rather than the full sequence $\pi$.

To improve efficiency, we incorporate the caching mechanism introduced in (Bergmeister et al., 2024). Once a coarsening sequence is generated, its levels are cached. During training, a random level is selected, returned, and then removed from the cache. A new sequence is generated only when the cache for a particular graph is depleted, avoiding unnecessary recomputation.

**Hyperparameters**. In all experiments described in Section 5, we use the following settings:

- Contraction family: The set of all edges in the clique representation, *i.e.*, $F(C) = \mathcal{E}$, for a weighted clique expansion $C = (\mathcal{V}, \mathcal{E})$.

- Cost function: Local Variation Cost (Loukas, 2019) with a preserving eigenspace size of $k = 8$.

- Reduction fraction range: $[\rho_{\min}, \rho_{\max}] = [0.1, 0.3]$.

---

**Algorithm 1 Hypergraph coarsening sequence sampling**: Randomized iterative coarsening of a hypergraph. At each step, contraction sets are selected based on cost, while ensuring right-side clusters never merge more than three at a time. Accepted contractions update the hypergraph structure, node scale encodings, and features.

---

1: **Parameters:** contraction family $F$, cost function $c$, reduction fraction range $[\rho_{\min}, \rho_{\max}]$, randomization parameter $\lambda$
2: **Input:** hypergraph $H$ with $n$ nodes and $m$ hyperedges; node features $\mathbf{F}_L$; hyperedge features $\mathbf{F}_R$
3: **Output:** coarsening sequence $\pi = (H^{(0)}, \dots, H^{(L)}) \in \Pi_F(H)$
4: Initialize $H^{(0)} \leftarrow H$
5: $B^{(0)} \leftarrow \text{BipartiteRepresentation}(H^{(0)})$
6: $C^{(0)} \leftarrow \text{WeightedCliqueExpansion}(H^{(0)})$
7: $\mathbf{b}_L^{(0)} \leftarrow (1, \dots, 1) \in \mathbb{R}^n$ {Initial node scale encodings}
8: $\mathbf{b}_R^{(0)} \leftarrow (1, \dots, 1) \in \mathbb{R}^m$ {Initial hyperedge scale encodings}
9: $\pi \leftarrow (B^{(0)}, \mathbf{b}_L^{(0)}, \mathbf{F}_L, \mathbf{F}_R)$
10: $l \leftarrow 0$
11: **while** $|\mathcal{V}_L^{(l)}| > 1$ **do**
12:     $l \leftarrow l + 1$
13:     red_frac $\sim \text{Uniform}([\rho_{\min}, \rho_{\max}])$ {Sample reduction fraction}
14:     $f \leftarrow c(\cdot, C^{(l-1)}, (\mathcal{P}^{(l-1)}, \dots, \mathcal{P}^{(0)}))$ {Cost function}
15:     accepted_contractions $\leftarrow \emptyset$
16:     **for** $S \in \text{SortedByCost}(F(C^{(l-1)}))$ **do**
17:         **if** $\text{Random}() > \lambda$ **then**
18:             **if** $S \cap (\bigcup_{P \in \text{accepted\_contractions}} P) = \emptyset$ **then**
19:                 $C_{\text{temp}} \leftarrow \text{CoarsenCliqueExpansion}(C^{(l-1)}, S)$
20:                 $B_{\text{temp}} \leftarrow \text{CoarsenBipartite}(B^{(l-1)}, S)$
21:             **if** $\forall$ right cluster $R \in B_{\text{temp}} : |R| \leq 3$ **then**
22:                 accepted_contractions $\leftarrow$ accepted_contractions $\cup \{S\}$
23:                 $C^{(l)} \leftarrow C_{\text{temp}}$
24:                 $B^{(l)} \leftarrow B_{\text{temp}}$
25:                 {*Update scale encodings and features for the new cluster*}
26:                 Let $S = \{v_1, \dots, v_k\}$ and the new node be $v^*$
27:                 $\mathbf{b}_L^{(l)}[v^*] \leftarrow \sum_{i=1}^k \mathbf{b}_L^{(l-1)}[v_i]$
28:                 $\mathbf{F}_L^{(l)}[v^*] \leftarrow \frac{1}{\mathbf{b}_L^{(l)}[v^*]} \sum_{i=1}^k \mathbf{b}_L^{(l-1)}[v_i] \cdot \mathbf{F}_L^{(l-1)}[v_i]$
29:             **end if**
30:             **end if**
31:         **end if**
32:         **if** $|\mathcal{V}_L^{(l-1)}| - |\bar{\mathcal{V}}_L^{(l)}| > \text{red\_frac} \cdot |\mathcal{V}_L^{(l-1)}|$ **then**
33:             **break**
34:         **end if**
35:     **end for**
36:     $\pi \leftarrow \pi \cup \{B^{(l)}, \mathbf{b}_L^{(l)}, \mathbf{F}_L^{(l)}\}$
37: **end while**
38: **return** $\pi$

---

# D. Examples of Coarsening Sequences

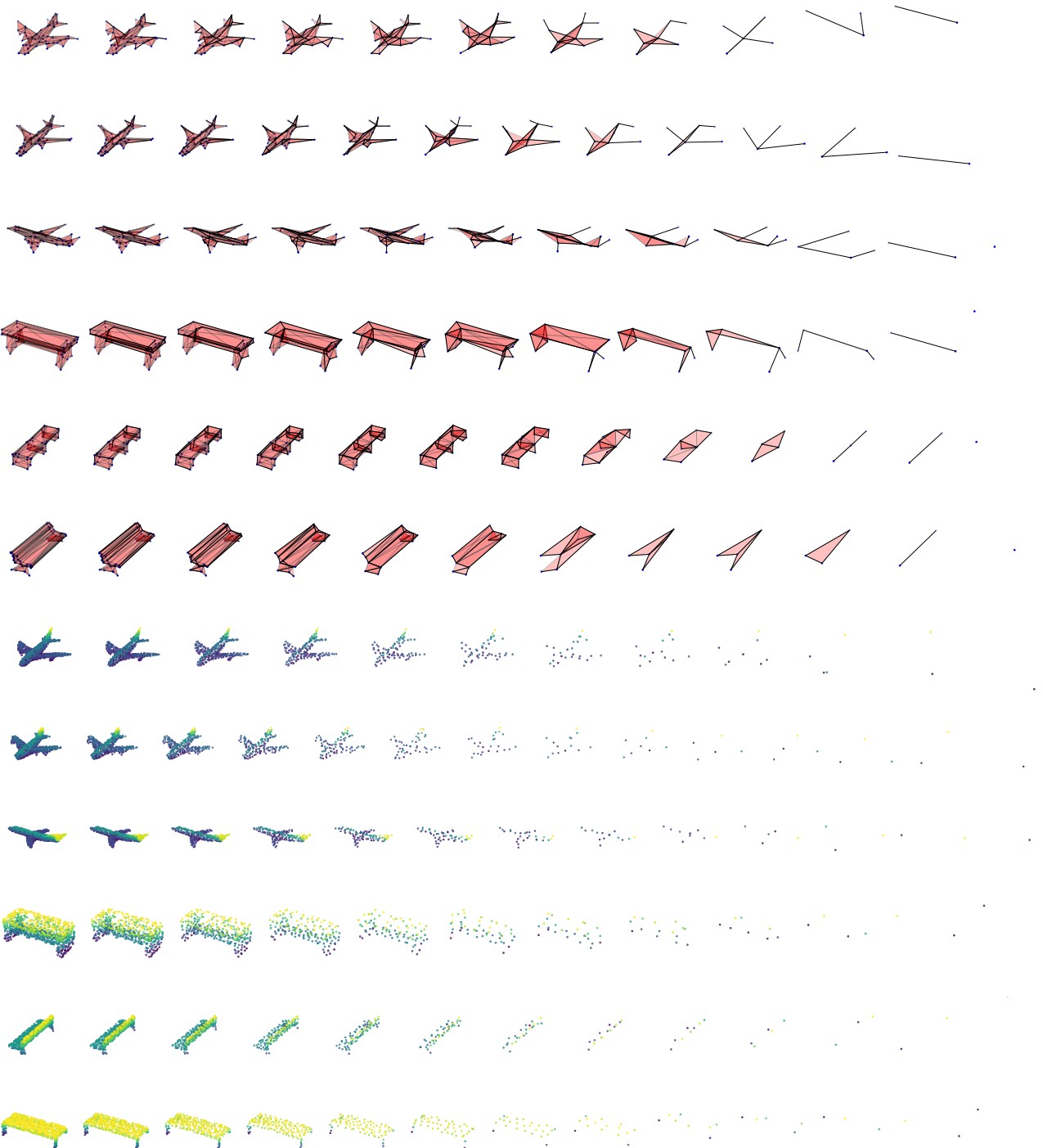

*Figure 3.* Examples of coarsening sequence for various meshes and point clouds. Thick lines represent 2-edges. Edges are omitted in point clouds for clarity.

# E. Experimental Details

In this section, we detail all three types of experiments – unfeatured hypergraphs, 3D meshes and graph point clouds – individually, detailing their datasets, baselines, metrics and specific hyperparameters.

For all experiments, we use embeddings with 32 dimensions for edge selection vectors and node and hyperedge expansion numbers. When they exist, features are embedded with 128 dimensions. SignNet always has 5 layers and a hidden dimension of 128. Positional encodings always have 32 dimensions. We always use 10 layers of Local PPGN. We use the AdamW (Loshchilov & Hutter, 2019) optimizer with a learning rate of $1e - 4$. All experiments are run for 1M steps on a single L40S. We use 8 CPU workers.

## E.1. Unfeatured Hypergraphs

**Datasets.** Our experiments use five datasets: three synthetic and two real-world, consistent with those described in (Gailhard et al., 2025):

- **Stochastic Block Model (SBM) hypergraphs** (Kim et al., 2018)**:** Constructed with 32 nodes split evenly into two groups. Every hyperedge connects three nodes. Hyperedges are sampled with probability 0.05 within groups and 0.001 between groups.
- **Ego hypergraphs** (Comrie & Kleinberg, 2021)**:** Created by generating an initial hypergraph of 150–200 nodes with 3000 randomly sampled hyperedges (up to 5 nodes each), then extracting an ego-centric subgraph by selecting a node and retaining only hyperedges that include it.
- **Tree-structured hypergraphs** (Nieminen & Peltola, 1999)**:** A tree with 32 nodes is generated using *networkx*, followed by grouping adjacent tree edges into hyperedges. Each hyperedge contains up to 5 nodes.
- **ModelNet40 meshes** (Wu et al., 2015)**:** Hypergraphs are derived from mesh topologies of selected ModelNet40 categories. To simplify computation, meshes are downsampled to fewer than 1000 vertices by iteratively merging nearby vertices. Duplicate triangles are removed, and the resulting low-poly mesh is converted into a hypergraph. We focus on the *bookshelf* and *piano* categories.

All datasets are divided into 128 training, 32 validation, and 40 testing hypergraphs.

**Evaluation Metrics.** We evaluate generated hypergraphs using the same suite of metrics as (Gailhard et al., 2025):

- **NodeNumDiff:** Average absolute difference in node count between generated and reference hypergraphs.
- **NodeDegreeDistrWasserstein:** Wasserstein distance between node degree distributions of generated and reference hypergraphs.
- **EdgeSizeDistrWasserstein:** Wasserstein distance between hyperedge size distributions.
- **Spectral:** Maximum Mean Discrepancy (MMD) between Laplacian spectra.
- **Uniqueness:** Fraction of generated hypergraphs that are non-isomorphic to one another.
- **Novelty:** Fraction of generated hypergraphs that are non-isomorphic to training samples.
- **CentralityCloseness, CentralityBetweenness, CentralityHarmonic:** Wasserstein distances computed between centrality distributions (on edges for $s = 1$). For details see (Aksoy et al., 2020).
- **ValidEgo:** For the *hypergraphEgo* dataset only, proportion of generated hypergraphs that contain a central node shared by all hyperedges.
- **ValidSBM:** For the *hypergraphSBM* dataset only, proportion of generated graphs that retain the original intra- and inter-group connectivity patterns.
- **ValidTree:** For the *hypergraphTree* dataset only, proportion of generated samples that preserve tree structure.

**Baselines.** We compare our method against the following baselines:

- **HyperPA** (Do et al., 2020): A heuristic approach for hypergraph generation.

- **Image-based models:** We design three baseline models—Diffusion, GAN, and VAE—that operate on incidence matrix representations of hypergraphs:

    - Each model is trained to produce binary images where white pixels signify node-hyperedge membership.
    - To normalize input sizes, incidence matrices are permuted randomly and padded with black pixels.
    - Generated images are thresholded to obtain binary incidence matrices.

- **HYGENE** (Gailhard et al., 2025): A hierarchical diffusion-based generator using reduction, expansion, and refinement steps.

**Specific hyperparameters.** We use $\lambda = 0.3$, our Local PPGN layers have a dimension of 128, and the hidden dimension for our MLP is 256. We use perturbed hypergraph expansion with a radius of 2 and dropout of 0.5. The SignNet relies on the top $K = 2$ eigenvalues and eigenvectors of the graph. We use 256 sampling steps during flow-matching. Our model has 4M parameters.

### E.2. 3D Meshes

**Datasets.** Datasets for meshes are taken from Manifold40 (Hu et al., 2022), which is a reworked version of ModelNet40 (Wu et al., 2015) to obtain manifold and watertight meshes. Meshes are subsequently coarsened to obtain low-poly versions of 50 vertices and 100 triangles. We use two classes:

- *Airplane* comprising 682 training samples, 21 validation samples, and 23 testing samples.
- *Bench* comprising 144 training samples, 19 validation samples, and 30 testing samples.

**Metrics.** We use the same metrics as for unfeatured hypergraphs. To this, we add *ChamDist*, which computes the minimal Chamfer distance between a point cloud sampled from the surface of the generated mesh and equivalent point clouds sampled from all validation/test set meshes, *ChamCov* (coverage), which measures the fraction of reference samples matched by at least one generated sample, and *ChamDiv* (diversity), which measures the average pairwise Chamfer distance between generated samples.

**Baselines.** We compare against a simple sequential baseline:

1. Our model (4M parameters) is trained for 1M steps on the topology of meshes **without** learning to generate the features.

2. A simple Local PPGN model (4M parameters) is trained for 20 epochs as a flow-matching model to learn to generate the 3D positions, with the topology fixed.

3. We use the best checkpoint of the first model to generate the topology, then apply the second model on this topology to generate the 3D positions.

Additionally, we compare our method to the following three simple baselines:

1. A flow-matching model of 6M parameters, using self-conditioning.

2. A Wasserstein GAN whose generator has 23M parameters and critic has 7M parameters.

3. A VAE with 23M parameters.

All models are trained for 100 epochs using the following framework:

- Each model is trained to produce binary images where white pixels signify node-hyperedge membership. An additional set of 3 dimension is concatenated to the row for each node, containing the position of the node.

- To normalize input sizes, incidence matrices are permuted randomly and padded with black pixels.

- Generated images are thresholded to obtain binary incidence matrices. Predicted positions are kept as is.

**Specific hyperparameters.** We use $\lambda = 0.1$, our Local PPGN layers have a dimension of 200, and the hidden dimension for our MLP is 300. We use perturbed hypergraph expansion with a radius of 2 and dropout of 0.5. The SignNet does not rely on the eigenvalues and eigenvectors of the graph. Instead, it takes as input a random feature tensor for each node, which is replicated across all expanded nodes belonging to the same cluster. We use 25 sampling steps during flow-matching. Our model has 6M parameters.

### E.3. Unfeatured Graph Datasets

**Datasets.** We reuse the same datasets used in (Bergmeister et al., 2024):

- **Planar graphs:** This dataset consists of 200 planar graphs, each containing 64 nodes. Graphs are obtained by applying Delaunay triangulation to points sampled uniformly at random within the unit square.
- **SBM graphs:** Comprising 200 samples, these graphs are generated from a stochastic block model with 2–5 communities. Each community contains 20–40 nodes. Edges are sampled with probability 0.3 between nodes in the same community and 0.05 otherwise.
- **Tree graphs:** We generate 200 random trees with 64 nodes using *networkx*.
- **Protein graphs:** (Dobson & Doig, 2003) provide a dataset where proteins are represented as graphs where nodes correspond to amino acids. An edge is introduced between two nodes if the Euclidean distance between the corresponding amino acids is less than 6 Å. Graph sizes vary considerably.
- **Point cloud graphs:** This dataset, introduced by (Liao et al., 2019), contains 41 point clouds representing household objects (Neumann et al., 2013). Each point cloud is converted into a graph, after which only the largest connected component is retained due to frequent disconnectedness in the raw graphs.

20% of samples are reserved for testing, while the remaining graphs are partitioned into 80% training and 20% validation.

**Evaluation Metrics.** We evaluate generated graphs using the same suite of metrics used in (Bergmeister et al., 2024). We compute the Maximum Mean Discrepancy (MMD) between generated and test graphs with respect to the following graph properties *Degree distribution*, *Clustering coefficient*, *Orbit counts*, *Spectrum*, *Wavelet coefficients*. To contextualize performance, we also compute these metrics between the training and test sets, and we report the mean ratio across all applicable metrics. Certain statistics are omitted for datasets where they are ill-defined or degenerate: for the *point cloud* dataset, the degree MMD is always zero due to the $k$-nearest-neighbor construction, and for the *tree* dataset, both clustering coefficients and orbit counts are identically zero. Finally, we include dataset-specific validity metrics:

- **ValidPlanar:** Proportion of generated graphs that remain planar.
- **ValidSBM:** Proportion consistent with the original SBM parameters.
- **ValidTree:** Proportion that preserve tree structure (*i.e.*, cycle-free).

**Baselines.** We compare against two state-of-the-art flat generative models, one graph bandwidth reduction method intended for scalability, and one hierarchical generative model: Defog (Qin et al., 2025), DiGress (Vignac et al., 2023), BwR (Diamant et al., 2023), and HSpectre (Bergmeister et al., 2024).

**Specific hyperparameters.** We use $\lambda = 0.3$, our Local PPGN layers have a dimension of 128, and the hidden dimension for our MLP is 256. We use perturbed hypergraph expansion with a radius of 2 and dropout of 0.5. The SignNet relies on the top $K = 2$ eigenvalues and eigenvectors of the graph for the *Planar*, *Point cloud* and *Tree* datasets, and none for the *SBM* and *Protein* datasets. For the *Point cloud* dataset, we use dropout with probability 0.1 during the computation of node embeddings and between each layer of our GNN. We use 256 sampling steps during flow-matching. Our model has 4M parameters.

### E.4. Graph Point Clouds

**Datasets.** We reuse the same datasets as for 3D meshes. Point clouds comprising 1024 nodes are sampled on the surface of each mesh, then each node is connected to its 3 nearest neighbors. Only the largest connected component is kept.

**Metrics.** We use the same metrics as for unfeatured graphs. To this, we add *ChamDist*, which computes the minimal Chamfer distance between a generated point cloud and all validation/test set point clouds, *ChamCov* (coverage), which

measures the fraction of reference point clouds matched by at least one generated sample, and *ChamDiv* (diversity), which measures the average pairwise Chamfer distance between generated point clouds.

**Baselines.** We compare against two state-of-the-art flat generative models: Defog (Qin et al., 2025), DiGress (Vignac et al., 2023). As these two methods are tailored for discrete data, we use a mixed-discrete continuous framework for continuous features.

**Specific hyperparameters.** We use $\lambda = 0.3$, our Local PPGN layers have a dimension of 128, and the hidden dimension for our MLP is 256. We use perturbed graph expansion with a radius of 2 and dropout of 0.5. The SignNet does not rely on the eigenvalues and eigenvectors of the graph. Instead, it takes as input a random feature tensor for each node, which is replicated across all expanded nodes belonging to the same cluster. We use 256 sampling steps during flow-matching. Our model has 4M parameters.

# F. Implementation Details

### F.1. Flow-matching Framework

We employ a flow-matching generative modeling framework (Lipman et al., 2023), with endpoint parameterization following (Dunn & Koes, 2024), equivalent to denoising diffusion models (Ho et al., 2020) using linear interpolation between the prior $p_0$ and target $p_1$ distributions. The goal is to align two distributions by transporting samples from $p_0$ to $p_1$ through a learned time-dependent vector field $\mathbf{f}(\mathbf{x}, t)$, governed by the ODE:

$$\frac{\partial \mathbf{x}}{\partial t} = \mathbf{f}(\mathbf{x}, t), \quad \mathbf{x}_0 \sim p_0. \tag{23}$$

Here, $\mathbf{f}(\mathbf{x}, t)$ is trained such that integrating this ODE produces samples from $p_1$.

**Endpoint parameterization.** Instead of directly modeling the flow field, we learn the terminal point $\hat{\mathbf{x}}_1(\mathbf{x}_t, 1)$ of the trajectory. The flow can be recovered using:

$$\mathbf{f}(\mathbf{x}_t, t) = \frac{\hat{\mathbf{x}}_1(\mathbf{x}_t, 1) - \mathbf{x}_t}{1 - t}. \tag{24}$$

**Training objective**. In our flow-matching framework, the model is trained to minimize the expected squared error between the predicted and true endpoints:

$$\mathcal{L} = \mathbb{E}_{t, \mathbf{x}_0 \sim p_0, \mathbf{x}_1 \sim p_1} \left[ \|\hat{\mathbf{x}}_1(\mathbf{x}_t, 1) - \mathbf{x}_1\|^2 \right], \tag{25}$$

where $\mathbf{x}_t = (1 - t)\mathbf{x}_0 + t\mathbf{x}_1$ is a linear interpolation between samples from the prior and target distributions. For the hypergraphs, 3D mesh and point cloud datasets, during training, $t$ is sampled using a log-normal distribution, which has been shown to improve performance (Esser et al., 2024). For regular unfeatured graph datasets, we use a standard uniform sampling.

In our setting, each loss term is masked so that it is evaluated only on the relevant nodes or hyperedges. Specifically, we retain: *(i)* expansion losses only for nodes with scale encoding greater than one ; *(ii)* scale encoding losses only for expanded nodes whose parent has a scale encoding greater than two; and *(iii)* feature losses only for expanded nodes. For a given sample, let

$$\mathcal{L}_{\text{node-expansion}} = \|\hat{\mathbf{v}}_L - \mathbf{v}_L\|^2 \, \mathbf{1}[\mathbf{b} > 1],$$

$$\mathcal{L}_{\text{scale encoding}} = \left\|\hat{\mathbf{f}} - \mathbf{f}\right\|^2 \, \mathbf{1}[\mathbf{b} > 2 \, \wedge \, \text{child of expanded cluster}],$$

$$\mathcal{L}_{\text{node-feature}} = \left\|\hat{\mathbf{F}}_L - \mathbf{F}_L^{\text{refine}}\right\|^2 \, \mathbf{1}[\text{child of expanded cluster}],$$

$$\mathcal{L}_{\text{hyperedge-expansion}} = \|\hat{\mathbf{v}}_R - \mathbf{v}_R\|^2,$$

$$\mathcal{L}_{\text{hyperedge-feature}} = \left\|\hat{\mathbf{F}}_R - \mathbf{F}_R^{\text{refine}}\right\|^2 \, \mathbf{1}[\text{child of expanded cluster}],$$

$$\mathcal{L}_{\text{incidence}} = \|\hat{\mathbf{e}} - \mathbf{e}\|^2,$$

where $\mathbf{1}[\cdot]$ denotes the indicator function. The total loss is the sum of all components, averaged across samples:

$$\mathcal{L} = \mathbb{E}[\mathcal{L}_{\text{node-expansion}} + \mathcal{L}_{\text{scale encoding}} + \mathcal{L}_{\text{node-feature}} + \mathcal{L}_{\text{hyperedge-expansion}} + \mathcal{L}_{\text{hyperedge-feature}} + \mathcal{L}_{\text{incidence}}].$$

**Graph inpainting**. During generation, we use inpainting to enforce constraints: *i)* scale encoding splits are fixed to 1 for unexpanded clusters or those of size 1, *ii)* equal splits are enforced for expanded clusters with size 2, *iii)* clusters of size one are not allowed to expand; and *iv)* features of non-expanded clusters are copied unchanged.

**Prior distributions and targets**. We use different prior distributions depending on the task:

- **Node and edge predictions:** Prior samples $p_0$ are drawn from a Gaussian distribution. Targets are either $-1$ or $1$, indicating binary decisions (*e.g.*, whether a node is expanded or an edge is retained).

- **Hyperedge expansion:** Similar to node and edge predictions, with targets $-1, 0$, or $1$, encoding the number of expansions (none, one, or two).

- **Scale encoding fractions:** Prior samples $p_0$ are drawn from a Dirichlet distribution with concentration parameter $\alpha = 1.5$, linearly mapped to $[-1, 1]$ via $2x - 1$. The target is the scale encoding fraction of each child node, linearly mapped to $[-1, 1]$ via $2x - 1$. Parent cluster scale encodings are encoded using sinusoidal positional encodings (Vaswani et al., 2017) (dimension 32, base frequency $10^{-4}$).

- **Feature generation:** Following (Ren et al., 2025), we draw prior features from a Gaussian and predict true node features, conditioned on the parent node's feature using a FiLM layer (Perez et al., 2018).

**Simplex projection via Von Neumann method**. When modeling scale encoding fractions, predictions must lie on the probability simplex. To ensure this, during generation, we project the model's outputs using the Von Neumann projection (Duchi et al., 2008), which finds the closest point (in Euclidean distance) on the simplex:

$$\Delta^K = \left\{ \mathbf{x} \in \mathbb{R}^K \mid x_i \geq 0, \sum_{i=1}^{K} x_i = 1 \right\}. \tag{26}$$

*i)* Sort $\mathbf{z} \in \mathbb{R}^K$ into a descending vector $\mathbf{u}$, such that $u_1 \geq u_2 \geq \cdots \geq u_K$.

*ii)* Find the smallest index $\rho \in \{1, \ldots, K\}$ such that:

$$u_\rho - \frac{1}{\rho} \left( \sum_{j=1}^{\rho} u_j - 1 \right) > 0. \tag{27}$$

*iii)* Compute the threshold:

$$\tau = \frac{1}{\rho} \left( \sum_{j=1}^{\rho} u_j - 1 \right). \tag{28}$$

*iv)* The projection is then:

$$\mathbf{x}^* = \max(\mathbf{z} - \tau, 0). \tag{29}$$

### F.2. Model Architecture

Our method represents the expansion numbers for left and right nodes, along with edge presence, as attributes of the bipartite graph. To model the distribution $p(\mathbf{v}_L^{(l)}, \mathbf{v}_R^{(l)}, \mathbf{e}^{(l)}, \mathbf{f}, \mathbf{F}_L^{\text{refine}} \mid \tilde{B}^{(l)})$, we adopt an endpoint-parameterized flow-matching framework (Lipman et al., 2023). Within this framework, the attributes—namely, the expansion vectors and edge indicators—are corrupted with noise, and a denoising network is trained to reconstruct the original values.

The denoising network is structured as follows:

1. **Positional encoding:** Node positions within the graph are encoded using SignNet (Lim et al., 2022). These encodings are replicated according to the respective expansion numbers.

2. **Attribute embedding:** Five separate linear layers are used to embed the bipartite graph attributes: left node features, right node features, edge features, node-specific features, and hyperedge-specific features. FiLM conditioning (Perez et al., 2018) is applied to incorporate contextual information into node and hyperedge features. Node scale encodings are embedded using sinusoidal positional encodings (Vaswani et al., 2017).

3. **Feature concatenation:**

   - For each left and right node, embeddings are concatenated with positional encodings and the desired reduction fraction. Left nodes also receive the node scale encoding embedding.
   - If node features are present, they are appended to the left nodes. Likewise, hyperedge features are appended to the right nodes when available.
   - For edges, embeddings include edge features, concatenated positional encodings of the incident nodes, and the reduction fraction.

4. **Graph processing:** The attribute-enriched bipartite graph is processed through a stack of sparse PPGN layers, following the architecture from (Bergmeister et al., 2024).

5. **Output prediction:** The final graph representations are passed through three linear projection heads to generate outputs.

   - Left node head: Predicts expansion values, scale encoding splits, and refined node features.
   - Right node head: Predicts hyperedge expansions and refined hyperedge features.
   - Edge head: Predicts edge existence.

### F.3. Additional Details

**Perturbed expansion.** Building on (Bergmeister et al., 2024; Gailhard et al., 2025), we augment Definitions 2 and 3—which are sufficient for reversing coarsening steps—with additional randomness to enhance generative quality. This modification is especially beneficial in low-data regimes where overfitting is a concern. Specifically, we introduce a probabilistic mechanism that supplements the set of edges $\tilde{\mathcal{E}}$ by randomly adding edges between node pairs on opposite sides of the bipartite graph that are within a fixed distance in $B$. The following definition extends the expansion process (Definition 2) to include this stochastic component.

**Definition 5** (Perturbed hypergraph expansion). Let $B = (\mathcal{V}_L, \mathcal{V}_R, \mathcal{E})$ be a bipartite graph, and let $\mathbf{v}_L \in \mathbb{N}^{|\mathcal{V}_L|}$ and $\mathbf{v}_R \in \mathbb{N}^{|\mathcal{V}_R|}$ denote the left and right cluster size vectors. For a given radius $r \in \mathbb{N}$ and probability $0 \leq p \leq 1$, we construct $\tilde{B}$ as in Definition 2. Additionally, for each pair of distinct nodes $\mathbf{v}_L(p) \in \tilde{\mathcal{V}}_L$ and $\mathbf{v}_R(q) \in \tilde{\mathcal{V}}_R$ that are within a distance of at most $2r + 1$ in $B$, we independently add an edge $e_{\{p_i, q_j\}}$ to $\tilde{\mathcal{E}}$ with probability $p$.

**Spectral conditioning.** In line with (Martinkus et al., 2022; Bergmeister et al., 2024), we incorporate spectral information—specifically, the principal eigenvalues and eigenvectors of the *normalized* Laplacian—as a form of conditioning during the generative process. This technique has been shown to improve the quality of generated graphs. To generate $B^{(l)}$ from its coarser form $B^{(l+1)}$, we leverage the approximate spectral invariance under coarsening. We compute the $k$ smallest non-zero eigenvalues and their corresponding eigenvectors from the normalized Laplacian matrix $\mathcal{L}^{(l+1)}$ of $B^{(l+1)}$. These eigenvectors are processed using SignNet (Lim et al., 2022) to produce node embeddings for $B^{(l+1)}$. These embeddings are then propagated to the expanded nodes of $B^{(l)}$, helping to preserve structural coherence and facilitate cluster identification. The hyperparameter $k$ controls the number of spectral components used.

**Multi-scale graph OT-coupling.** Minibatch OT-coupling (Tong et al., 2024; Pooladian et al., 2023) accelerates training by jointly sampling priors and targets and reindexing priors to minimize input–output distance, leading to smoother flows and fewer inference steps. As detailed in Section 3.5, we adapt this idea by treating each bipartite graph as a minibatch and applying OT-coupling within them. To preserve the distribution, we restrict permutations to groups of *equivalent nodes* (identical topology and features, differing only in noise). For efficiency, we assume equivalence only among nodes from the same cluster expansion. Since each cluster produces two or three children, coupling reduces to evaluating two or six possible permutations per cluster, which can be efficiently parallelized with tensors (Algorithm 2).

---

**Algorithm 2** Multi-scale graph OT coupling for coarsening/expansion

---

1: **Input:** Expanded bipartite representation $B$ with clusters $\{V^{(p)}\}$, each $V^{(p)}$ containing 1 or 2 nodes; target samples $\{x_i\}$
2: **Output:** Reindexed noise samples $\{\tilde{z}_i\}$
3: Sample noise $\{z_i\}$ for each node in $B$
4: **for all** clusters $V^{(p)} \in B$ **do**
5:     **if** $|V^{(p)}| = 1$ **then**
6:         No reassignment needed for singleton cluster
7:     **else if** $|V^{(p)}| = 2$ **then**
8:         Let $i, j$ be the indices of the two nodes in $V^{(p)}$
9:         Let $x_i, x_j$ be their corresponding targets
10:        Compute normal order cost:
$$C_{\text{normal}} \leftarrow \|z_i - x_i\|^2 + \|z_j - x_j\|^2$$
11:        Compute swapped order cost:
$$C_{\text{swap}} \leftarrow \|z_j - x_i\|^2 + \|z_i - x_j\|^2$$
12:        **if** $C_{\text{swap}} < C_{\text{normal}}$ **then**
13:           Swap $z_i \leftrightarrow z_j$
14:        **end if**
15:     **end if**
16: **end for**
17: **return** $\{\tilde{z}_i\}$ (reassigned noise samples)

---

# G. Training and Sampling Procedures

In this section, we present the complete training and inference procedures, detailed in Algorithms 4 and 5. Both pipelines rely on node embeddings produced by Algorithm 3.

---

**Algorithm 3 Node embedding computation:** Compute left and right node embeddings for a bipartite representation of a hypergraph. Embeddings are expanded according to cluster size vectors.

---

1: **Parameters:** number of spectral features $k$
2: **Input:** bipartite representation $B = (\mathcal{V}_L, \mathcal{V}_R, \mathcal{E})$, spectral feature model $\text{SignNet}_\theta$, cluster size vectors $\mathbf{v}_L$ and $\mathbf{v}_R$
3: **Output:** node embeddings computed for all nodes in $\mathcal{V}_L$ and $\mathcal{V}_R$, replicated according to $\mathbf{v}_L$ and $\mathbf{v}_R$
4: **if** $k = 0$ **then**
5:     Sample random embeddings:

$$\mathbf{H} = [h^{(1)}, \ldots, h^{(|\mathcal{V}|)}] \overset{i.i.d.}{\sim} \mathcal{N}(0, I)$$

6: **else**
7:     **if** $k < |\mathcal{V}|$ **then**
8:         Compute $k$ spectral features:

$$[\lambda_1, \ldots, \lambda_k], [u_1, \ldots, u_k] \leftarrow \text{EIG}(B)$$

9:     **else**
10:        Compute all spectral features and pad with zeros:

$$[\lambda_1, \ldots, \lambda_{|\mathcal{V}_L| + |\mathcal{V}_R| - 1}], [u_1, \ldots, u_{|\mathcal{V}_L| + |\mathcal{V}_R| - 1}] \leftarrow \text{EIG}(B)$$

$$[\lambda_{|\mathcal{V}_L| + |\mathcal{V}_R|}, \ldots, \lambda_k], [u_{|\mathcal{V}_L| + |\mathcal{V}_R|}, \ldots, u_k] \leftarrow [0, \ldots, 0], [0, \ldots, 0]$$

11:     **end if**
12:     Compute embeddings with SignNet:

$$\mathbf{H} = [h^{(1)}, \ldots, h^{(|\mathcal{V}_L| + |\mathcal{V}_R|)}] \leftarrow \text{SignNet}_\theta([\lambda_1, \ldots, \lambda_k], [u_1, \ldots, u_k], B)$$

13: **end if**
14: Expand bipartite representation according to cluster sizes:

$$\tilde{B} = (\mathcal{V}_L^{(1)} \cup \cdots \cup \mathcal{V}_L^{(p_l)}, \mathcal{V}_R^{(1)} \cup \cdots \cup \mathcal{V}_R^{(p_r)}, \tilde{\mathcal{E}}) \leftarrow \tilde{B}(B, \mathbf{v}_L, \mathbf{v}_R)$$

15: Replicate embeddings according to cluster sizes: For all $p_L \in [|\mathcal{V}_L|]$ and $p_R \in [|\mathcal{V}_R|]$:

$$\tilde{\mathbf{H}}[p_i] = \mathbf{H}[p_l], \quad \forall \mathbf{v}_L^{(p_i)} \in \mathcal{V}_L^{(p_l)}$$

$$\tilde{\mathbf{H}}[p_i] = \mathbf{H}[p_r], \quad \forall \mathbf{v}_R^{(p_i)} \in \mathcal{V}_R^{(p_r)}$$

16: **return** $\tilde{\mathbf{H}}$

---

---

**Algorithm 4 End-to-end training procedure:** Complete training procedure for our model.

---

1: **Parameters:** number of spectral features $k$ for node embeddings
2: **Input:** dataset $\mathcal{D} = \{H_1, \ldots, H_N\}$, denoising model $\text{GNN}_\theta$, spectral feature model $\text{SignNet}_\theta$
3: **Output:** trained model parameters $\theta$
4: **while** not converged **do**
5:    Sample a graph $H \sim \text{Uniform}(\mathcal{D})$
6:    Sample coarsening sequence:

$$(B^{(0)}, \ldots, B^{(L)}) \leftarrow \text{RndRedSeq}(H) \text{ (Algorithm 1)}$$

7:    Sample level $l \sim \text{Uniform}(\{0, \ldots, L\})$
8:    **if** $l = 0$ **then**
9:        Set $\mathbf{v}_L^{(0)} \leftarrow 1, \mathbf{v}_R^{(0)} \leftarrow 1$
10:   **else**
11:       Set $\mathbf{v}_L^{(l)}, \mathbf{v}_R^{(l)}$ such that $\tilde{B}(B^{(l)}, \mathbf{v}_L^{(l)}, \mathbf{v}_R^{(l)})$ equals the node set of $B^{(l-1)}$
12:   **end if**
13:   **if** $l = L$ **then**
14:       Initialize last-level bipartite graph:

$$B^{(l+1)} \leftarrow B^{(l)} = (\{1\}, \{2\}, \{(1,2)\}, \mathbf{b} = size(H), \mathbf{F}_L = 0, \mathbf{F}_R = 0)$$

15:       $\mathbf{v}_L^{(l+1)} \leftarrow 1, \mathbf{v}_R^{(l+1)} \leftarrow 1, \mathbf{e}^{(l)} \leftarrow 1$
16:   **end if**
17:   Set $\mathbf{e}^{(l)}, \mathbf{f}, \mathbf{F}_L^{\text{refine}}, \mathbf{F}_R^{\text{refine}}$ such that

$$B(\tilde{B}(B^{(l+1)}, \mathbf{v}_L^{(l+1)}, \mathbf{v}_R^{(l+1)}), \mathbf{e}^{(l)}, \mathbf{f}, \mathbf{F}_L^{\text{refine}}, \mathbf{F}_R^{\text{refine}}) = B^{(L)}$$

18:   Compute node embeddings:

$$\mathbf{H}^{(l)} \leftarrow \text{Embeddings}(B^{(l+1)}, \text{SignNet}_\theta, \mathbf{v}_L^{(l+1)}, \mathbf{v}_R^{(l+1)})$$

19:   Compute reduction fraction:

$$\hat{\rho} \leftarrow 1 - \frac{n^{(l)}}{n^{(l-1)}}$$

where $n^{(l)}, n^{(l-1)}$ are the sizes of the left side of $B^{(l)}$ and $B^{(l-1)}$
20:   Compute denoising output:
$$D_\theta \leftarrow \text{GNN}_\theta(\cdot, \cdot, \tilde{B}^{(l)}, \mathbf{H}^{(l)}, n^{(0)}, \rho)$$

where $n^{(0)}$ is the size of the left side of $B^{(0)}$
21:   Take gradient descent step on $\nabla_\theta \text{DiffusionLoss}(\mathbf{v}_L^{(L)}, \mathbf{v}_R^{(L)}, \mathbf{e}^{(l)}, \mathbf{f}, \mathbf{F}_L^{\text{refine}}, \mathbf{F}_R^{\text{refine}}, D_\theta)$
22: **end while**
23: **return** $\theta$

---

---

**Algorithm 5 End-to-end sampling procedure with deterministic expansion size:** This describes the sampling procedure. Assumes maximum cluster sizes 2 and 3.

---

1: **Parameters:** reduction fraction range $[\rho_{\min}, \rho_{\max}]$
2: **Input:** target hypergraph size $N$, denoising model $\text{GNN}_\theta$, spectral feature model $\text{SignNet}_\theta$
3: **Output:** sampled hypergraph $H = (\mathcal{V}, \mathcal{E})$ with $|\mathcal{V}| = N$
4: Initialize minimal bipartite graph:

$$B = (\mathcal{V}_L, \mathcal{V}_R, \mathcal{E}, \mathbf{f}, \mathbf{F}_L^{\text{refine}}, \mathbf{F}_R^{\text{refine}}) \leftarrow (\{1\}, \{2\}, \{(1, 2)\}, N, 0, 0)$$

5: Initialize cluster size vectors: $\mathbf{v}_L \leftarrow [1]$, $\mathbf{v}_R \leftarrow [1]$
6: **while** $|\mathcal{V}_L| < N$ **do**
7:     Compute node embeddings:

$$\mathbf{H} \leftarrow \text{Embeddings}(B, \text{SignNet}_\theta, \mathbf{v}_L, \mathbf{v}_R)$$

8:     $n \leftarrow \|\mathbf{v}_L\|_1$
9:     Sample reduction fraction: $\rho \sim \text{Uniform}([\rho_{\min}, \rho_{\max}])$
10:    Compute number of left-side nodes to add:

$$n^+ \leftarrow \lceil \rho(n + n^+) \rceil, \quad n^+ \leftarrow \min(n^+, N - n)$$

11:    Actual reduction fraction: $\hat{\rho} \leftarrow 1 - \frac{n}{n + n^+}$
12:    Compute denoising output:

$$D_\theta \leftarrow \text{GNN}_\theta(\cdot, \cdot, \tilde{B}(B, \mathbf{v}_L, \mathbf{v}_R), \mathbf{H}, N, \hat{\rho})$$

13:    Sample features:

$$(\mathbf{v}_L)_0, (\mathbf{v}_R)_0, (\mathbf{e})_0, \mathbf{f}, \mathbf{F}_L^{\text{refine}}, \mathbf{F}_R^{\text{refine}} \leftarrow \text{Sample}(D_\theta)$$

14:    Update cluster size vectors:

$$\mathbf{v}_L[i] = \begin{cases} 2 & \text{if } |\{j \mid (\mathbf{v}_L)_0[j] \geq (\mathbf{v}_L)_0[i]\}| \geq n^+ \\ 1 & \text{otherwise} \end{cases}$$

$$\mathbf{v}_R[i] = \begin{cases} 1 & (\mathbf{v}_R)_0[i] < 1.66 \\ 2 & 1.66 \leq (\mathbf{v}_R)_0[i] < 2.33 \\ 3 & \text{otherwise} \end{cases}$$

15:    Update edges:

$$\mathbf{e}[i] = \begin{cases} 1 & (\mathbf{e})_0[i] > 0.5 \\ 0 & \text{otherwise} \end{cases}$$

16:    Refine bipartite graph:

$$B = (\mathcal{V}_L, \mathcal{V}_R, \mathcal{E}) \leftarrow B(\tilde{B}, \mathbf{e}, \mathbf{f}, \mathbf{F}_L^{\text{refine}}, \mathbf{F}_R^{\text{refine}})$$

17: **end while**
18: Build hypergraph $H$ from its bipartite representation $B$
19: **return** $H$

---

# H. Complexity Analysis

In this section, we investigate the asymptotic complexity of our proposed algorithm, which extends the methodology introduced by (Bergmeister et al., 2024) and (Gailhard et al., 2025). To construct a hypergraph comprising $n$ nodes, $m$ hyperedges, and $k$ incidences, the algorithm sequentially produces a series of bipartite graphs $B^{(L)} = (\{1\}, \{2\}, \{(1,2)\}), B^{(L-1)}, \ldots, B^{(0)} = B$, where the final graph $B$ corresponds to the bipartite representation of the generated hypergraph. We use $n$, $m$, and $k$ to denote, respectively, the number of nodes, hyperedges, and incidences in the hypergraph, and as the number of left-side nodes, right-side nodes, and edges in the corresponding bipartite graph.

For each level $0 \leq l < L$ of the sequence, the number of left-side nodes in $B^{(l)}$, denoted $n_l$, satisfies $n_l \geq (1+\epsilon)n_{l-1}$ for some $\epsilon > 0$ (*e.g.*, $\epsilon = \texttt{reduction\_frac}/(1 - \texttt{reduction\_frac})$). This implies an upper bound on the number of steps in the expansion sequence: $\lceil \log_{1+\epsilon} n \rceil \in \mathcal{O}(\log n)$. Since the expansion process only increases node counts, all $B_l$ graphs contain fewer than $n$ left-side and $m$ right-side nodes. The number of edges, however, may temporarily exceed $k$, as the intermediate bipartite graphs may include additional edges removed in later refinements. Still, because the coarsening during training consistently reduces incidences, the model is expected to learn accurate edge refinement and avoid such accumulation. Consequently, we assume $k_l \leq k$ and $m_l \leq m$ for all $0 \leq l \leq L$.

Next, we assess the computational cost of generating a single expansion step. At level $l = L$, this consists of creating a pair of connected nodes, initializing features as matrices of zeros, initializing scale encoding as the targeted node count, and predicting the expansion vectors $\mathbf{v}_L$ and $\mathbf{v}_R$—a process with constant complexity $\mathcal{O}(1)$. For levels $0 \leq l < L$, given $B^{(l+1)}$ and expansion vectors $\mathbf{v}_L^{(l+1)}$, $\mathbf{v}_R^{(l+1)}$, the algorithm constructs the expanded bipartite graph $\tilde{B}(B^{(l+1)}, \mathbf{v}_L^{(l+1)}, \mathbf{v}_R^{(l+1)})$ in $\mathcal{O}(n+m)$ time. It then samples $\mathbf{v}_L^{(l)}$, $\mathbf{v}_R^{(l)}$, $\mathbf{e}^{(l)}$, $\mathbf{f}$, $\mathbf{F}_L^{\text{refine}}$, and $\mathbf{F}_R^{\text{refine}}$, and constructs the refined graph $B^{(l)} = B(\tilde{B}^{(l)}, \mathbf{e}^{(l)}, \mathbf{f}, \mathbf{F}_L^{\text{refine}}, \mathbf{F}_R^{\text{refine}})$. Letting $v_{\max}^L$ and $v_{\max}^R$ be the maximum cluster sizes, the incidence count in $\tilde{B}^{(l)}$ is bounded by $k_l \leq k_{l+1} v_{\max}^L v_{\max}^R$.

The sampling process queries a denoising model a constant number of times per step. The complexity is thus governed by the architecture. In our case, since bipartite graphs are triangle-free, the *Local PPGN* model (Bergmeister et al., 2024) has linear complexity $\mathcal{O}(n + m + k)$. Embedding computation for $B^{(l)}$ similarly costs $\mathcal{O}(n + m + k)$. This includes calculating the top $K$ eigenvalues/eigenvectors of the Laplacian via the method from (Vishnoi, 2013), with complexity $\mathcal{O}(K(n_{l+1} + m_{l+1} + k_{l+1}))$, and embedding via *SignNet*, also linear in graph size due to fixed $K$.

The final transformation from the bipartite graph to a hypergraph—by collapsing right-side nodes into hyperedges—has a cost of $\mathcal{O}(m + k)$. Under these assumptions, the total complexity to generate a hypergraph $H$ with $n$ nodes, $m$ hyperedges, and $k$ incidences is $\mathcal{O}((n + m + k) \log n)$.

# I. Detailed Numerical Results

In this section, we present detailed numerical results for all datasets and metrics described in Appendix E. Reported values of the form $a \pm b$ indicate the mean $a$ and twice the standard deviation $b$ computed over 5 runs. The best and second-best results are highlighted in **bold** and underlined, respectively.

## I.1. Comparisons with the Baselines

*Table 6.* Detailed numerical results for unfeatured hypergraphs.

| **SBM Hypergraphs** | | | | | | | | |
|---|---|---|---|---|---|---|---|---|
| **Method** | **Valid (%) ↑** | **NumDiff ↓** | **Deg ↓** | **EdgeSize ↓** | **Spectral ↓** | **Harmonic ↓** | **Closeness ↓** | **Betweenness ↓** |
| HyperPA (Do et al., 2020) | 2.5 | 0.075 | 4.062 | 0.407 | 0.273 | 77.840 | 0.074 | 0.008 |
| VAE (Kingma & Welling, 2014) | 0.0 | 0.375 | 1.280 | 1.059 | 0.024 | 6.543 | **0.007** | 0.006 |
| GAN (Goodfellow et al., 2020) | 0.0 | 1.200 | 2.106 | 1.203 | 0.059 | 10.700 | 0.076 | 0.012 |
| Diffusion (Ho et al., 2020) | 0.0 | 0.150 | 1.717 | 1.390 | 0.031 | 13.940 | 0.040 | 0.004 |
| HYGENE (Gailhard et al., 2025) | 65.0 | 0.525 | **0.321** | **0.002** | 0.010 | **2.990** | 0.016 | **0.000** |
| FAHNES | **87.8±3.1** | **0.029±0.009** | 0.846±0.457 | 0.005±0.003 | **0.006±0.004** | 6.410±3.124 | 0.009±0.006 | 0.003±0.001 |

| **Ego Hypergraphs** | | | | | | | | |
|---|---|---|---|---|---|---|---|---|
| **Method** | **Valid (%) ↑** | **NumDiff ↓** | **Deg ↓** | **EdgeSize ↓** | **Spectral ↓** | **Harmonic ↓** | **Closeness ↓** | **Betweenness ↓** |
| HyperPA (Do et al., 2020) | 0.0 | 35.830 | 2.590 | 0.423 | 0.237 | 143.000 | 0.354 | 0.002 |
| VAE (Kingma & Welling, 2014) | 0.0 | 47.580 | 0.803 | 1.458 | 0.133 | 38.950 | 0.558 | 0.019 |
| GAN (Goodfellow et al., 2020) | 0.0 | 60.350 | 0.917 | 1.665 | 0.230 | 41.800 | 0.612 | 0.015 |
| Diffusion (Ho et al., 2020) | 0.0 | 4.475 | 3.984 | 2.985 | 0.190 | 6.911 | 0.407 | 0.009 |
| HYGENE (Gailhard et al., 2025) | 90.0 | 12.550 | **0.063** | 0.220 | **0.004** | 5.790 | 0.025 | **0.000** |
| FAHNES | **99.5±1.1** | **0.128±0.171** | 0.124±0.086 | **0.155±0.067** | **0.004±0.003** | **2.703±3.468** | **0.003±0.005** | **0.000±0.000** |

| **Tree Hypergraphs** | | | | | | | | |
|---|---|---|---|---|---|---|---|---|
| **Method** | **Valid (%) ↑** | **NumDiff ↓** | **Deg ↓** | **EdgeSize ↓** | **Spectral ↓** | **Harmonic ↓** | **Closeness ↓** | **Betweenness ↓** |
| HyperPA (Do et al., 2020) | 0.0 | 2.350 | 0.315 | 0.284 | 0.159 | 5.941 | 0.477 | 0.168 |
| VAE (Kingma & Welling, 2014) | 0.0 | 9.700 | 0.072 | 0.480 | 0.124 | 3.869 | 0.280 | 0.139 |
| GAN (Goodfellow et al., 2020) | 0.0 | 6.000 | 0.151 | 0.469 | 0.089 | 2.198 | 0.201 | 0.124 |
| Diffusion (Ho et al., 2020) | 0.0 | 2.225 | 1.718 | 1.922 | 0.127 | 8.565 | 0.353 | 0.139 |
| HYGENE (Gailhard et al., 2025) | 77.5 | **0.000** | 0.059 | 0.108 | 0.012 | 1.099 | 0.041 | 0.016 |
| FAHNES | **89.7±6.0** | **0.000±0.000** | **0.022±0.022** | **0.030±0.034** | **0.003±0.002** | **0.171±0.106** | **0.014±0.006** | **0.014±0.004** |

| **ModelNet Bookshelf** | | | | | | | |
|---|---|---|---|---|---|---|---|
| **Method** | **NumDiff ↓** | **Deg ↓** | **EdgeSize ↓** | **Spectral ↓** | **Harmonic ↓** | **Closeness ↓** | **Betweenness ↓** |
| HyperPA (Do et al., 2020) | 8.025 | 7.562 | 0.044 | 0.048 | 877.500 | 0.211 | 0.005 |
| VAE (Kingma & Welling, 2014) | 47.450 | 6.190 | 1.520 | 0.190 | 113.600 | 0.145 | 0.003 |
| GAN (Goodfellow et al., 2020) | **0.000** | 397.200 | 46.300 | 0.476 | 670.100 | 0.707 | 0.007 |
| Diffusion (Ho et al., 2020) | **0.000** | 20.360 | 2.346 | 0.079 | 264.100 | 0.239 | 0.006 |
| HYGENE (Gailhard et al., 2025) | 69.730 | **1.050** | 0.034 | 0.068 | **27.400** | 0.204 | 0.004 |
| FAHNES | 0.135±0.276 | 2.980±2.107 | **0.020±0.025** | **0.024±0.015** | 46.614±58.803 | 0.086±0.030 | **0.001±0.001** |

| **ModelNet Piano** | | | | | | | |
|---|---|---|---|---|---|---|---|
| **Method** | **NumDiff ↓** | **Deg ↓** | **EdgeSize ↓** | **Spectral ↓** | **Harmonic ↓** | **Closeness ↓** | **Betweenness ↓** |
| HyperPA (Do et al., 2020) | 0.825 | 9.254 | **0.023** | 0.067 | **77.840** | 0.236 | 0.004 |
| VAE (Kingma & Welling, 2014) | 75.350 | 8.060 | 1.686 | 0.396 | 184.300 | 0.241 | 0.003 |
| GAN (Goodfellow et al., 2020) | **0.000** | 409.000 | 86.380 | 0.697 | 622.200 | 0.738 | 0.005 |
| Diffusion (Ho et al., 2020) | 0.050 | 20.900 | 4.192 | 0.113 | 289.300 | 0.303 | 0.004 |
| HYGENE (Gailhard et al., 2025) | 42.520 | 6.290 | 0.027 | 0.117 | 155.000 | 0.285 | **0.002** |
| FAHNES | 0.846±1.009 | **3.265±1.954** | 0.042±0.056 | **0.040±0.026** | 119.158±81.023 | **0.123±0.119** | 0.002±0.002 |

*Table 7.* Detailed numerical results for 3D meshes.

| | | | | ManifoldNet Airplane | | | | | | |
|---|---|---|---|---|---|---|---|---|---|---|
| Method | ChamDist ↓ | ChamCov (%) ↑ | ChamDiv ↑ | NumDiff ↓ | Deg ↓ | EdgeSize ↓ | Spectral ↓ | Harmonic ↓ | Closeness ↓ | Betweenness ↓ |
| Sequential | 0.082±0.074 | **34.8±0.0** | 0.129±0.011 | 0.116±0.074 | 0.258±0.009 | 0.010±0.004 | **0.010±0.001** | 4.491±0.663 | 0.058±0.005 | 0.005±0.000 |
| WGAN | 0.124±0.021 | 10.4±3.5 | 0.037±0.016 | 1.435±0.000 | 68.128±14.602 | 36.138±7.739 | 0.614±0.071 | 57.422±0.000 | 0.655±0.000 | 0.020±0.000 |
| Flow Matching | 0.085±0.014 | 20.9±10.0 | 0.063±0.005 | 0.574±0.216 | 0.620±0.130 | 1.089±0.052 | 0.040±0.004 | 10.782±3.903 | 0.156±0.028 | 0.009±0.001 |
| VAE | 0.231±0.208 | 26.9±11.7 | **0.191±0.130** | 1.603±1.340 | 35.488±70.110 | 18.896±35.756 | 0.328±0.588 | 36.716±38.851 | 0.425±0.431 | 0.015±0.007 |
| FAHNES | **0.048±0.003** | 31.9±13.2 | 0.101±0.011 | **0.017±0.070** | 0.218±0.085 | **0.004±0.007** | 0.010±0.003 | 2.428±1.455 | 0.032±0.007 | 0.003±0.001 |
| | | | | **ManifoldNet Bench** | | | | | | |
| Method | ChamDist ↓ | ChamCov (%) ↑ | ChamDiv ↑ | NumDiff ↓ | Deg ↓ | EdgeSize ↓ | Spectral ↓ | Harmonic ↓ | Closeness ↓ | Betweenness ↓ |
| Sequential | 0.169±0.007 | 13.3±0.0 | 0.157±0.003 | **0.022±0.031** | 0.340±0.052 | 0.004±0.003 | **0.007±0.000** | 3.694±0.510 | 0.046±0.000 | 0.004±0.000 |
| WGAN | 0.128±0.015 | 17.3±3.3 | 0.063±0.099 | 2.267±0.000 | 69.748±20.988 | 36.352±10.914 | 0.621±0.066 | 56.108±0.000 | 0.640±0.000 | 0.019±0.000 |
| Flow Matching | 0.108±0.019 | 30.7±2.5 | 0.112±0.007 | 2.327±0.265 | 0.774±0.112 | 1.142±0.047 | 0.038±0.006 | 9.608±4.260 | 0.145±0.033 | 0.008±0.002 |
| VAE | 0.280±0.223 | 27.5±9.9 | 0.195±0.107 | 1.853±1.309 | 24.476±65.177 | 13.207±33.338 | 0.228±0.557 | 31.415±35.421 | 0.361±0.397 | 0.014±0.007 |
| FAHNES | **0.064±0.005** | **42.2±13.9** | **0.243±0.037** | 0.060±0.149 | 0.349±0.454 | **0.009±0.017** | 0.017±0.011 | 5.069±8.171 | 0.046±0.040 | 0.004±0.003 |

*Table 8.* Detailed numerical results for unfeatured graph datasets.

| | | | SBM Graphs | | | |
|---|---|---|---|---|---|---|
| Method | Valid (%) ↑ | Wavelet ↓ | Orbit ↓ | Clustering ↓ | Deg ↓ | Spectral ↓ | Ratio ↓ |
| DiGress (Vignac et al., 2023) | 60.0 | **0.001** | 0.042 | **0.049** | 0.002 | **0.004** | **1.7** |
| DeFoG (Qin et al., 2025) | **90.0±5.1** | 0.008±0.002 | 0.056±0.074 | 0.052±0.001 | **0.001±0.002** | 0.005±0.001 | 4.9±1.3 |
| HSpectre (Bergmeister et al., 2024) | 45.0 | 0.022 | 0.067 | 0.052 | 0.012 | 0.007 | 10.2 |
| BwR (Diamant et al., 2023) | 7.5 | 0.089 | 0.114 | 0.064 | 0.048 | 0.0169 | 38.6 |
| FAHNES | 50.0±5.0 | 0.008±0.003 | **0.040±0.000** | 0.052±0.004 | 0.007±0.000 | **0.004±0.002** | 4.8±0.7 |
| | | | **Tree Graphs** | | | |
| Method | Valid (%) ↑ | Wavelet ↓ | Orbit ↓ | Clustering ↓ | Deg ↓ | Spectral ↓ | Ratio ↓ |
| DiGress (Vignac et al., 2023) | 90.0 | **0.004** | **0.000** | **0.000** | **0.000** | **0.011** | **1.6** |
| DeFoG (Qin et al., 2025) | 96.5±2.6 | 0.005±0.000 | 0.000±0.000 | 0.000±0.000 | 0.000±0.000 | 0.011±0.003 | 1.6±0.4 |
| HSpectre (Bergmeister et al., 2024) | **100.0** | 0.005 | **0.000** | **0.000** | **0.000** | 0.012 | 4.0 |
| BwR (Diamant et al., 2023) | 0.0 | 0.039 | 0.000 | 0.124 | 0.002 | 0.048 | 11.4 |
| FAHNES | **100.0±0.0** | 0.005±0.001 | **0.000±0.000** | **0.000±0.000** | **0.000±0.000** | 0.012±0.004 | 1.8±0.8 |
| | | | **Planar Graphs** | | | |
| Method | Valid (%) ↑ | Wavelet ↓ | Orbit ↓ | Clustering ↓ | Deg ↓ | Spectral ↓ | Ratio ↓ |
| DiGress (Vignac et al., 2023) | 77.5 | 0.003 | 0.008 | 0.078 | 0.001 | 0.010 | 5.1 |
| DeFoG (Qin et al., 2025) | **99.5±1.0** | **0.001±0.000** | **0.001±0.000** | 0.050±0.015 | 0.001±0.000 | 0.007±0.001 | **1.6±0.4** |
| HSpectre (Bergmeister et al., 2024) | 95.0 | **0.001** | 0.002 | 0.063 | 0.001 | 0.007 | 2.1 |
| BwR (Diamant et al., 2023) | 0.0 | 0.131 | 0.547 | 0.260 | 0.023 | 0.044 | 251.9 |
| FAHNES | 96.7±6.2 | **0.001±0.001** | 0.003±0.005 | **0.042±0.002** | **0.000±0.000** | 0.008±0.003 | 2.2±1.6 |
| | | | **Protein Graphs** | | | |
| Method | | Wavelet ↓ | Orbit ↓ | Clustering ↓ | Deg ↓ | Spectral ↓ | Ratio ↓ |
| DiGress (Vignac et al., 2023) | | 0.006 | 0.129 | 0.049 | 0.004 | 0.002 | 18.0 |
| HSpectre (Bergmeister et al., 2024) | | 0.003 | **0.005** | **0.031** | 0.003 | **0.001** | 5.9 |
| BwR (Diamant et al., 2023) | | 0.120 | 0.494 | 0.420 | 0.126 | 0.070 | 245.4 |
| FAHNES | | **0.001±0.001** | 0.013±0.004 | 0.039±0.005 | **0.000±0.001** | **0.001±0.001** | **3.8±1.9** |
| | | | **Point Cloud (Unfeatured) Graphs** | | | |
| Method | | Wavelet ↓ | Orbit ↓ | Clustering ↓ | Deg ↓ | Spectral ↓ | Ratio ↓ |
| DiGress (Vignac et al., 2023) | | OOM | OOM | OOM | OOM | OOM | OOM |
| HSpectre (Bergmeister et al., 2024) | | 0.019 | 0.078 | 0.578 | 0.014 | **0.005** | 7.0 |
| BwR (Diamant et al., 2023) | | 0.592 | 1.073 | 0.469 | 0.493 | 0.291 | 133.2 |
| FAHNES | | **0.018±0.005** | **0.009±0.001** | **0.441±0.073** | **0.006±0.012** | 0.006±0.001 | **3.2±0.5** |

*Table 9.* Detailed numerical results for graph point cloud datasets.

| | | | | **Airplane Point Clouds** | | | | | | |
|---|---|---|---|---|---|---|---|---|---|---|
| Method | ChamDist ↓ | ChamCov (%) ↑ | ChamDiv ↑ | NumDiff ↓ | Wavelet ↓ | Orbit ↓ | Clustering ↓ | Deg ↓ | Spectral ↓ | Ratio ↓ |
| DiGress (Vignac et al., 2023) | OOM | OOM | OOM | OOM | OOM | OOM | OOM | OOM | OOM | OOM |
| DeFoG (Qin et al., 2025) | OOM | OOM | OOM | OOM | OOM | OOM | OOM | OOM | OOM | OOM |
| FAHNES | $0.094 \pm 0.006$ | $22.2 \pm 14.5$ | $0.085 \pm 0.026$ | $0.000 \pm 0.000$ | $0.004 \pm 0.001$ | $0.074 \pm 0.104$ | $0.264 \pm 0.254$ | $0.002 \pm 0.002$ | $0.005 \pm 0.004$ | $67.3 \pm 44.9$ |
| | | | | **Bench Point Clouds** | | | | | | |
| Method | ChamDist ↓ | ChamCov (%) ↑ | ChamDiv ↑ | NumDiff ↓ | Wavelet ↓ | Orbit ↓ | Clustering ↓ | Deg ↓ | Spectral ↓ | Ratio ↓ |
| DiGress (Vignac et al., 2023) | OOM | OOM | OOM | OOM | OOM | OOM | OOM | OOM | OOM | OOM |
| DeFoG (Qin et al., 2025) | OOM | OOM | OOM | OOM | OOM | OOM | OOM | OOM | OOM | OOM |
| FAHNES | $0.130 \pm 0.001$ | $15.8 \pm 10.6$ | $0.191 \pm 0.114$ | $0.000 \pm 0.000$ | $0.003 \pm 0.000$ | $0.013 \pm 0.004$ | $0.229 \pm 0.071$ | $0.000 \pm 0.000$ | $0.004 \pm 0.000$ | $73.3 \pm 22.2$ |

## I.2. Ablation Studies

*Table 10.* Detailed numerical results for ablation studies on unfeatured hypergraphs.

| | | | | **SBM Hypergraphs** | | | | | |
|---|---|---|---|---|---|---|---|---|---|
| Scale Encoding | Graph OT Coupling | Valid (%) ↑ | NumDiff ↓ | Deg ↓ | EdgeSize ↓ | Spectral ↓ | Harmonic ↓ | Closeness ↓ | Betweenness ↓ |
| ✓ | ✓ | $87.8 \pm 3.1$ | $0.029 \pm 0.009$ | $0.846 \pm 0.457$ | $0.005 \pm 0.003$ | $0.006 \pm 0.004$ | $6.410 \pm 3.124$ | $0.009 \pm 0.006$ | $0.003 \pm 0.001$ |
| ✗ | ✓ | $85.3 \pm 5.9$ | $0.044 \pm 0.078$ | $0.856 \pm 0.636$ | $0.023 \pm 0.022$ | $0.006 \pm 0.004$ | $6.210 \pm 4.649$ | $0.009 \pm 0.013$ | $0.003 \pm 0.003$ |
| ✓ | ✗ | $86.7 \pm 6.7$ | $0.039 \pm 0.014$ | $0.910 \pm 0.363$ | $0.006 \pm 0.004$ | $0.006 \pm 0.004$ | $6.815 \pm 1.913$ | $0.010 \pm 0.005$ | $0.004 \pm 0.001$ |
| ✗ | ✗ | $84.6 \pm 4.6$ | $0.049 \pm 0.045$ | $0.916 \pm 0.749$ | $0.047 \pm 0.070$ | $0.007 \pm 0.007$ | $6.665 \pm 5.199$ | $0.012 \pm 0.014$ | $0.004 \pm 0.004$ |
| | | | | **Ego Hypergraphs** | | | | | |
| Scale Encoding | Graph OT Coupling | Valid (%) ↑ | NumDiff ↓ | Deg ↓ | EdgeSize ↓ | Spectral ↓ | Harmonic ↓ | Closeness ↓ | Betweenness ↓ |
| ✓ | ✓ | $99.5 \pm 1.1$ | $0.128 \pm 0.171$ | $0.124 \pm 0.086$ | $0.155 \pm 0.067$ | $0.004 \pm 0.003$ | $2.703 \pm 3.468$ | $0.003 \pm 0.005$ | $0.000 \pm 0.000$ |
| ✗ | ✓ | $100.0 \pm 0.0$ | $0.118 \pm 0.158$ | $0.140 \pm 0.104$ | $0.133 \pm 0.106$ | $0.005 \pm 0.003$ | $3.426 \pm 2.629$ | $0.000 \pm 0.000$ | $0.000 \pm 0.000$ |
| ✓ | ✗ | $99.9 \pm 0.4$ | $0.073 \pm 0.050$ | $0.237 \pm 0.465$ | $0.193 \pm 0.175$ | $0.007 \pm 0.012$ | $4.521 \pm 5.629$ | $0.000 \pm 0.002$ | $0.000 \pm 0.000$ |
| ✗ | ✗ | $99.5 \pm 1.1$ | $0.117 \pm 0.155$ | $0.110 \pm 0.111$ | $0.189 \pm 0.084$ | $0.004 \pm 0.003$ | $2.765 \pm 1.684$ | $0.001 \pm 0.004$ | $0.000 \pm 0.000$ |
| | | | | **Tree Hypergraphs** | | | | | |
| Scale Encoding | Graph OT Coupling | Valid (%) ↑ | NumDiff ↓ | Deg ↓ | EdgeSize ↓ | Spectral ↓ | Harmonic ↓ | Closeness ↓ | Betweenness ↓ |
| ✓ | ✓ | $89.7 \pm 6.0$ | $0.000 \pm 0.000$ | $0.022 \pm 0.022$ | $0.030 \pm 0.034$ | $0.003 \pm 0.002$ | $0.171 \pm 0.106$ | $0.014 \pm 0.006$ | $0.014 \pm 0.004$ |
| ✗ | ✓ | $90.7 \pm 6.7$ | $0.000 \pm 0.000$ | $0.024 \pm 0.019$ | $0.067 \pm 0.016$ | $0.004 \pm 0.001$ | $0.315 \pm 0.098$ | $0.018 \pm 0.012$ | $0.014 \pm 0.008$ |
| ✓ | ✗ | $89.3 \pm 3.6$ | $0.000 \pm 0.000$ | $0.019 \pm 0.013$ | $0.047 \pm 0.077$ | $0.005 \pm 0.011$ | $0.169 \pm 0.106$ | $0.041 \pm 0.082$ | $0.013 \pm 0.010$ |
| ✗ | ✗ | $95.6 \pm 3.7$ | $0.000 \pm 0.000$ | $0.042 \pm 0.042$ | $0.077 \pm 0.091$ | $0.005 \pm 0.003$ | $0.217 \pm 0.169$ | $0.013 \pm 0.013$ | $0.008 \pm 0.011$ |
| | | | | **ModelNet Bookshelf** | | | | | |
| Scale Encoding | Graph OT Coupling | | NumDiff ↓ | Deg ↓ | EdgeSize ↓ | Spectral ↓ | Harmonic ↓ | Closeness ↓ | Betweenness ↓ |
| ✓ | ✓ | | $0.135 \pm 0.276$ | $2.980 \pm 2.107$ | $0.020 \pm 0.025$ | $0.024 \pm 0.015$ | $46.614 \pm 58.803$ | $0.086 \pm 0.030$ | $0.001 \pm 0.001$ |
| ✗ | ✓ | | $0.940 \pm 0.917$ | $3.040 \pm 1.446$ | $0.089 \pm 0.099$ | $0.032 \pm 0.013$ | $59.300 \pm 108.101$ | $0.137 \pm 0.042$ | $0.003 \pm 0.001$ |
| ✓ | ✗ | | $0.265 \pm 0.496$ | $4.400 \pm 1.635$ | $0.025 \pm 0.021$ | $0.014 \pm 0.007$ | $54.558 \pm 30.276$ | $0.102 \pm 0.020$ | $0.001 \pm 0.001$ |
| ✗ | ✗ | | $1.325 \pm 1.631$ | $2.820 \pm 0.958$ | $0.055 \pm 0.077$ | $0.031 \pm 0.009$ | $79.889 \pm 98.721$ | $0.135 \pm 0.012$ | $0.003 \pm 0.001$ |
| | | | | **ModelNet Piano** | | | | | |
| Scale Encoding | Graph OT Coupling | | NumDiff ↓ | Deg ↓ | EdgeSize ↓ | Spectral ↓ | Harmonic ↓ | Closeness ↓ | Betweenness ↓ |
| ✓ | ✓ | | $0.846 \pm 1.009$ | $3.265 \pm 1.954$ | $0.042 \pm 0.056$ | $0.040 \pm 0.026$ | $119.158 \pm 81.023$ | $0.123 \pm 0.119$ | $0.002 \pm 0.002$ |
| ✗ | ✓ | | $3.622 \pm 1.822$ | $3.358 \pm 1.772$ | $0.054 \pm 0.080$ | $0.055 \pm 0.040$ | $133.806 \pm 137.603$ | $0.155 \pm 0.058$ | $0.012 \pm 0.033$ |
| ✓ | ✗ | | $3.155 \pm 3.637$ | $5.250 \pm 1.087$ | $0.066 \pm 0.168$ | $0.030 \pm 0.048$ | $154.211 \pm 383.979$ | $0.188 \pm 0.164$ | $0.002 \pm 0.002$ |
| ✗ | ✗ | | $5.490 \pm 8.847$ | $3.733 \pm 2.989$ | $0.107 \pm 0.215$ | $0.036 \pm 0.016$ | $97.724 \pm 198.774$ | $0.138 \pm 0.129$ | $0.002 \pm 0.002$ |

*Table 11.* Detailed numerical results for ablation studies on 3D meshes.

| | | **ManifoldNet Airplane** | | | | | | | | | |
|---|---|---|---|---|---|---|---|---|---|---|---|
| Scale Encoding | Graph OT Coupling | ChamDist ↓ | ChamCov (%) ↑ | ChamDiv ↑ | NumDiff ↓ | Deg ↓ | EdgeSize ↓ | Spectral ↓ | Harmonic ↓ | Closeness ↓ | Betweenness ↓ |
| ✓ | ✓ | **0.048**±0.003 | 31.9±13.2 | 0.101±0.011 | **0.017**±0.070 | **0.218**±0.085 | **0.004**±0.007 | 0.010±0.003 | 2.428±1.455 | 0.032±0.007 | **0.003**±0.001 |
| ✗ | ✓ | 0.079±0.019 | 29.0±17.6 | 0.080±0.011 | 0.426±0.820 | 0.588±0.451 | 0.024±0.021 | 0.014±0.007 | 2.429±1.394 | 0.033±0.028 | **0.003**±0.002 |
| ✓ | ✗ | 0.050±0.005 | 30.4±15.0 | 0.093±0.019 | 0.052±0.065 | 0.235±0.063 | 0.014±0.027 | 0.012±0.005 | **1.604**±1.400 | 0.022±0.007 | **0.003**±0.001 |
| ✗ | ✗ | 0.100±0.023 | **44.9**±10.0 | **0.114**±0.031 | 0.304±0.437 | 0.864±0.358 | 0.020±0.016 | 0.019±0.009 | 4.184±2.176 | 0.025±0.009 | 0.005±0.002 |

| | | **ManifoldNet Bench** | | | | | | | | | |
|---|---|---|---|---|---|---|---|---|---|---|---|
| Scale Encoding | Graph OT Coupling | ChamDist ↓ | ChamCov (%) ↑ | ChamDiv ↑ | NumDiff ↓ | Deg ↓ | EdgeSize ↓ | Spectral ↓ | Harmonic ↓ | Closeness ↓ | Betweenness ↓ |
| ✓ | ✓ | **0.064**±0.005 | **42.2**±13.9 | **0.243**±0.037 | 0.060±0.149 | 0.349±0.454 | 0.009±0.017 | 0.017±0.011 | 5.069±8.171 | 0.046±0.040 | **0.004**±0.003 |
| ✗ | ✓ | 0.090±0.003 | 37.8±16.8 | 0.232±0.048 | 0.240±0.265 | 1.089±0.346 | 0.013±0.015 | 0.018±0.009 | 6.507±2.771 | **0.014**±0.005 | 0.005±0.002 |
| ✓ | ✗ | 0.085±0.056 | 32.2±7.7 | 0.208±0.030 | **0.020**±0.032 | **0.221**±0.160 | **0.006**±0.004 | **0.013**±0.003 | **1.864**±2.828 | 0.028±0.019 | 0.005±0.006 |
| ✗ | ✗ | 0.098±0.024 | **42.2**±22.0 | 0.235±0.053 | 0.267±0.319 | 1.242±0.582 | 0.013±0.011 | 0.022±0.014 | 7.619±4.165 | 0.015±0.016 | 0.006±0.004 |

We additionally provide the following results, where we try different strategies for the coarsening process:

1. *Random* corresponds to random mergings of nodes, where the local variation cost in Algorithm 1 is replaced by a random value.

2. *Feature* corresponds to mergings that prioritize pairs of adjacent nodes with similar features, where the local variation cost in Algorithm 1 is replaced by the squared Euclidean distance between features.

Results are shown in Table 12.

*Table 12.* Ablations on the coarsening strategy employed.

| **ManifoldNet Bench** | | | | | |
|---|---|---|---|---|---|
| Method | ChamDist ↓ | NumDiff ↓ | Deg ↓ | EdgeSize ↓ | Spectral ↓ |
| Random Mergings | 0.158 | 2.300 | **0.324** | **0.002** | 0.019 |
| Feature-Preserving | 0.085 | 0.200 | 0.750 | 0.004 | 0.030 |
| Spectrum-Preserving | **0.064** | **0.060** | 0.349 | 0.009 | **0.017** |

| **ManifoldNet Airplane** | | | | | |
|---|---|---|---|---|---|
| Method | ChamDist ↓ | NumDiff ↓ | Deg ↓ | EdgeSize ↓ | Spectral ↓ |
| Random Mergings | 0.054 | **0.000** | 0.423 | 0.009 | 0.011 |
| Feature-Preserving | 0.056 | **0.000** | 0.241 | **0.002** | 0.014 |
| Spectrum-Preserving | **0.048** | 0.017 | **0.218** | 0.004 | **0.010** |

## J. Comparison between Training and Generated Samples

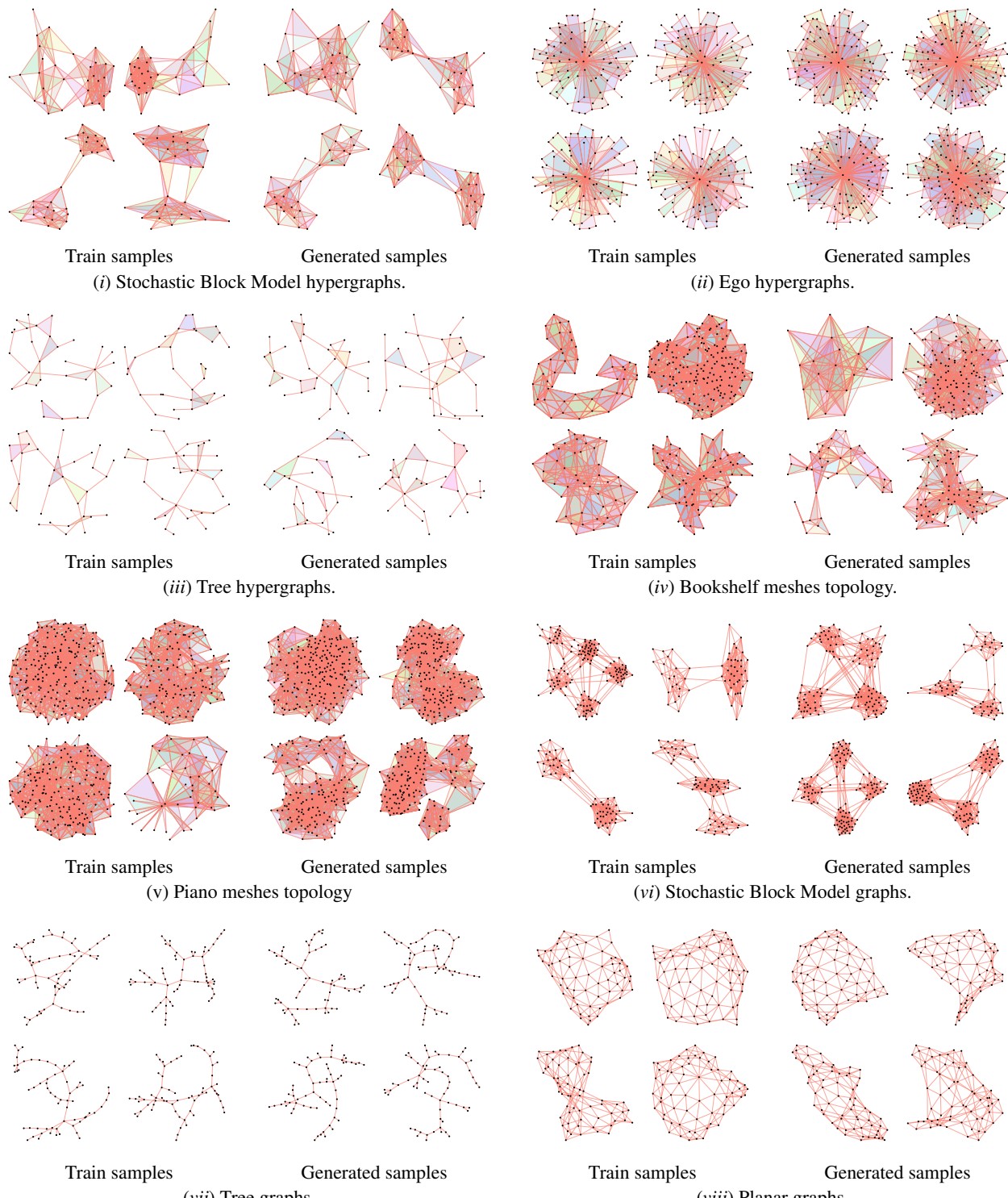

Train samples      Generated samples

(*i*) Stochastic Block Model hypergraphs.

Train samples      Generated samples

(*ii*) Ego hypergraphs.

Train samples      Generated samples

(*iii*) Tree hypergraphs.

Train samples      Generated samples

(*iv*) Bookshelf meshes topology.

Train samples      Generated samples

(v) Piano meshes topology

Train samples      Generated samples

(*vi*) Stochastic Block Model graphs.

Train samples      Generated samples

(*vii*) Tree graphs.

Train samples      Generated samples

(*viii*) Planar graphs.

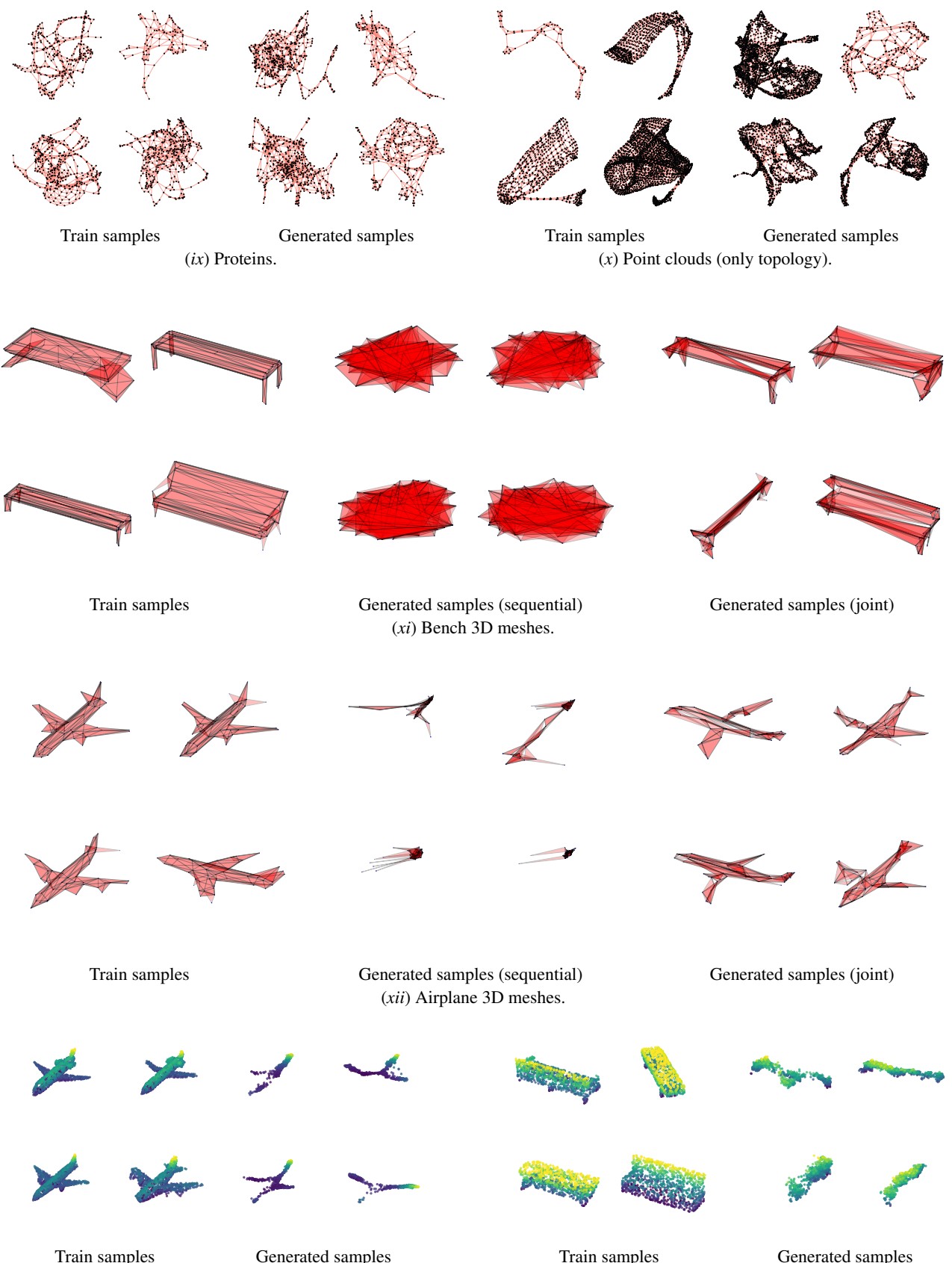

Train samples      Generated samples      Train samples      Generated samples

(*ix*) Proteins.      (*x*) Point clouds (only topology).

Train samples      Generated samples (sequential)      Generated samples (joint)

(*xi*) Bench 3D meshes.

Train samples      Generated samples (sequential)      Generated samples (joint)

(*xii*) Airplane 3D meshes.

Train samples      Generated samples      Train samples      Generated samples

(*xiii*) ManifoldNet Airplane point clouds.      (*xiv*) ManifoldNet Bench point clouds.

