# OpenReview forum: "Feature-Aware (Hyper)graph Generation via Next-Scale Prediction"
_ICML.cc/2026/Conference — ICML 2026 regular_

### Official Review · Reviewer_TGqy · 2026-03-11

**Soundness:** 3
**Presentation:** 3
**Significance:** 4
**Originality:** 4
**Overall Recommendation:** 4
**Confidence:** 4

**Summary:**

The paper tackles a notoriously difficult problem in geometric deep learning: the scalable generation of complex, large-scale (hyper)graphs with rich node and edge features. While traditional graph generative models excel on small molecules or simple topologies, they suffer from severe scalability bottlenecks and often fail to capture higher-order relationships (hyperedges) alongside continuous/discrete features. To address this, the authors propose FAHNES, a hierarchical framework that frames the generative process as "next-scale prediction." By employing multi-scale representations through iterative node coarsening and localized expansion, the model effectively learns the structural and feature distributions from a coarse, macroscopic level down to the fine-grained, microscopic details. This joint generation of both topology and features across different resolutions presents a highly generalized approach to complex network synthesis.

**Compliance With Llm Reviewing Policy:**

Affirmed.

**Final Justification:**

In my initial review, I highlighted the strong originality and significance of the proposed "next-scale prediction" framework. The conceptual shift to hierarchical, multi-scale modeling offers an elegant solution to the scalability bottlenecks in joint topology and feature generation for complex hypergraphs. My primary reservations centered around the soundness of the methodology in extreme cases—specifically, the potential biases introduced by the graph coarsening strategy, the optimization stability when balancing discrete and continuous modalities, and the comprehensiveness of the hypergraph evaluation metrics.

The authors provided a constructive rebuttal that adequately addressed my main concerns. Their clarifications regarding the robustness of the coarsening process and the optimization dynamics improved my confidence in the technical soundness of the work. While evaluating higher-order topological invariants remains an open challenge for the broader community, the authors have sufficiently justified their current empirical approach.

Weighing all dimensions, the paper's originality and its significant contribution to scaling geometric deep learning strongly outweigh its minor limitations. The methodology is well-motivated, and the presentation is clear. The rebuttal has reinforced my prior positive assessment, and I am maintaining my recommendation of Weak Accept.

**Key Questions For Authors:**

How sensitive is the quality of the generated (hyper)graphs to the specific coarsening algorithm used to construct the multi-scale hierarchy? Have you observed scenarios where the coarsening process irreversibly destroys critical topological information (e.g., in star-like or bipartite structures)?

Could you elaborate on the optimization dynamics when jointly learning topology and features? How do you weigh the loss components for discrete structural prediction versus feature prediction, and does the model require careful hyperparameter tuning to avoid mode collapse in one of the modalities?

Regarding the evaluation metrics for hypergraphs: do the current statistics evaluate the complex intersection patterns of the hyperedges directly, or are they computed on the projected pairwise graphs? (Clarifying this would significantly strengthen the empirical claims).

**Limitations:**

The authors should explicitly discuss the computational overhead introduced by the multi-scale hierarchy construction during the training phase. Additionally, a critical limitation that needs addressing is the framework's applicability to networks with extreme structural heterophily, where the basic assumption of semantic similarity within coarsened super-nodes might break down. There are no obvious negative societal impacts associated with this foundational algorithmic work.

**Strengths And Weaknesses:**

Strengths And Weaknesses

Strengths:

The core conceptual shift to "next-scale prediction" is highly compelling. By explicitly modeling the hierarchical and multi-scale nature of real-world networks, the authors elegantly bypass the quadratic or exponential complexity limits that plague flat, one-shot, or autoregressive generative models on large graphs.

The framework’s versatility is a major asset. Designing a model that can jointly generate topology and features is already challenging; extending this capability to hypergraphs—which inherently contain complex, non-pairwise interactions—demonstrates a strong mastery of geometric representation learning.

The methodology for local expansion from coarsened nodes is well-motivated. It intuitively mirrors the real-world evolution of complex systems (e.g., social communities dividing, biological structures differentiating), making the generative process not just mathematically sound, but also structurally meaningful.

Weaknesses:

The performance and structural fidelity of the generated graphs seem heavily dependent on the chosen graph coarsening strategy during the forward (encoding) process. The paper does not fully explore how different algebraic or spectral coarsening techniques might bias the generative distribution, especially for scale-free or highly disassortative networks where naive coarsening might destroy critical hub structures.

Jointly generating discrete topological structures (edges/hyperedges) and features (which might be continuous) typically introduces severe training instabilities or loss-balancing issues. The manuscript could be more transparent about the optimization challenges and how the framework prevents one modality (e.g., node features) from dominating the learning signal over the other.

Evaluating hypergraph generation remains an open challenge in the community. While the empirical results are positive, it is somewhat unclear if the chosen metrics fully capture the preservation of true higher-order topological invariants, or if they primarily reflect lower-order (pairwise clique-expansion) statistics.

---

> ### Author Rebuttal · Authors · 2026-03-30
>
> We thank the reviewer for the thoughtful comments. We will incorporate these points and our responses into the potential camera-ready version.
>
> **W1)** Indeed, coarsening is an important component of our pipeline. We include an ablation on the coarsening algorithm in Table 12 (Appendix I), where we try three different strategies: spectrum-preserving, feature-preserving, and random. We found spectrum-preserving coarsening to be the best choice for our experiments, similarly to [2,3].
>
> That said, the choice of coarsening sequence is outside the main scope of this paper and is left to future work.
>
> **W2, Q2)** In practice, we did not observe any loss imbalance between topology and features during training. The only standard preprocessing we applied was centering and normalizing meshes and point clouds to unit variance.
>
> We remind that topological predictions are generated continuously, not via discrete flow matching: a Gaussian prior is mapped to the interval [-1, 1], where -1 and 1 mean no expansion and expansion (or edge deletion or conservation), and then discretized.
>
> The challenges we did encounter were different:
> - Unstable targets, which we addressed using OT coupling,
> - Mismatches in scale across graph regions, solved by our scale encoding, and
> - Vanishing training signals for later scales, analyzed in Proposition 4.
>
> No special hyperparameter tuning was needed. Please note that our additional components—scale encoding and OT-coupling—do not introduce new hyperparameters.
>
> **W3, Q3)** Indeed, evaluating graph and hypergraph generative models is still an open research question [1], so we reused the evaluation framework of [2, 3] for consistency. All metrics for unfeatured hypergraphs are computed directly on the hypergraphs, not on any proxy graph such as a clique expansion (see Appendix E1).
> In particular:
> - We measure the Wasserstein distance between the hyperedge size distributions of predicted and reference samples.
> - We compute the Maximum Mean Discrepancy (MMD) between the spectra of predicted and reference hypergraphs, using Bolla’s Laplacian [4], which is specific to hypergraphs.
>
> These metrics explicitly capture higher-order structural properties, rather than relying on pairwise projections.
>
> For 3D meshes and point clouds, visual inspection clearly demonstrates that our method produces meaningful structures, as shown in Appendix J.
>
> **Q1)** Expansion/refinement inverts any coarsening sequence (see Appendix A.3 of [3]), so in principle, coarsening cannot irreversibly destroy topological information. The main limitation is the expressivity of the model, which must predict the correct inputs for expansion/refinement. We use a local PPGN model [2], which is more expressive than standard GNNs and well-suited to this hierarchical strategy. Empirically, spectrum-preserving coarsening seems to help preserve star-like structures (see Table 3 in [3] for an ablation on Ego structures).
>
> Regarding sensitivity to the coarsening choice, we include an ablation on different strategies: spectrum-preserving, feature-preserving, and random (Table 12, Appendix I). For the rebuttal, we additionally ran a sensitivity analysis on the *rand_lambda* parameter for spectrum-preserving coarsening; please refer to answer to **W2)** of reviewer **2Way**.
>
> **Limitations)** During training, the continuous generation of coarsening sequences does introduce some overhead. However, it is fully CPU-based and can be efficiently parallelized with multiple workers while training on GPU (straightforward in PyTorch), resulting in minimal impact on overall training time when adequate CPU resources are available.
>
> Regarding heterophily, our method does not rely on homophily assumptions. Coarsening is performed on the spectral structure of the graph, independent of node features [5]. Moreover, flow matching is highly expressive and can learn complex mappings between nodes and features, even when neighboring nodes are semantically dissimilar.
> Thus, while extreme heterophily may change the patterns learned, there is no fundamental limitation in the framework preventing it from modeling such graphs. In fact, our method obtains satisfying results for the two 3D mesh datasets, comprising heterophilic hypergraphs—wings for airplanes and legs for benches are made of hyperedges connecting nodes with very dissimilar positions.
>
> We hope the responses have addressed your concerns, and we will modify the paper accordingly in our final version. If you have further suggestions, we are also happy to integrate them.
>
> ---
>
> [1] Evaluation Metrics for Graph Generative Models: Problems, Pitfalls, and Practical Solutions, ICLR 2022.
>
> [2] Efficient and Scalable Graph Generation, ICLR 2024.
>
> [3] HYGENE: A Diffusion-based Hypergraph Generation Method, AAAI 2025.
>
> [4] Spectra, Euclidean Representations and Clusterings of Hypergraphs, Discrete Mathematics, 1993.
>
> [5] Graph Reduction with Spectral and Cut Guarantees, JMLR, 2019.

---

> > ### Author Rebuttal · Reviewer_TGqy · 2026-04-02
> >
> > Thank you for your response and clarification. I acknowledge your explanation, although my overall assessment remains unchanged.

---

### Official Review · Reviewer_oSXS · 2026-03-12

**Soundness:** 3
**Presentation:** 3
**Significance:** 3
**Originality:** 3
**Overall Recommendation:** 4
**Confidence:** 4

**Summary:**

The paper presents FAHNES, a hierarchical generative framework designed to jointly generate the topology and features of both graphs and hypergraphs. Addressing the scalability limitations of previous methods and their frequent neglect of node/edge features, FAHNES employs a next-scale prediction approach. It first coarsens the structure into a multi-scale representation and then learns to reverse this process via localized expansion and refinement steps. Key technical contributions include a hierarchical scale encoding to maintain structural consistency across scales and a multi-scale graph Optimal Transport (OT) coupling to align generated structures with ground truth during training, handling permutation invariance efficiently. The method is validated on synthetic hypergraphs, 3D meshes, and point cloud datasets, demonstrating state-of-the-art performance.

**Compliance With Llm Reviewing Policy:**

Affirmed.

**Final Justification:**

The authors addressed my concerns and I will keep my positive score.

**Key Questions For Authors:**

1. How difficult would it be to adapt the flow matching and refinement steps to handle categorical features (e.g., using discrete diffusion or categorical flow matching)?

**Limitations:**

As stated in the paper, the framework explicitly assumes continuous feature distributions (Gaussian priors) and uses continuous flow matching. However, a vast majority of graph generation tasks (especially in chemistry and biology, like molecule generation) involve categorical features (atom types, bond types). The inability to handle discrete features directly is a practical limitation compared to models like DiGress or discrete diffusion methods.

**Strengths And Weaknesses:**

### Strengths
1. The model offers a generalized approach that works effectively for both graphs and hypergraphs (via bipartite representation), which is a significant contribution given that hypergraph generation is less explored.
2. Unlike many hierarchical models that focus primarily on topology, FAHNES successfully integrates feature generation (e.g., 3D coordinates) at every scale, which is crucial for real-world applications like 3D shape modeling.
3. The method demonstrates superior performance across a wide range of datasets, from synthetic benchmarks to complex real-world data like 3D meshes and point clouds, significantly outperforming existing baselines (e.g., HYGENE, DiGress).

### Weaknesses
1. The generative process is strictly the inverse of the coarsening process. If the spectrum-preserving coarsening (or any chosen algorithm) destroys critical structural information that cannot be easily recovered, the generation quality will suffer. The paper does not thoroughly analyze the sensitivity of the model to the choice of coarsening algorithm, effectively treating it as a "black box" pre-processing step.
2. The pipeline is highly complex (coarsening + bipartite mapping + scale encoding + OT coupling + flow matching). For smaller or simpler graphs, simpler flat models might achieve comparable performance with much lower implementation complexity. The paper lacks a detailed analysis of where this complexity pays off versus where it is overkill.

---

> ### Author Rebuttal · Authors · 2026-03-30
>
> We thank the reviewer for the thoughtful comments. We will incorporate these points and our responses into the potential camera-ready version.
>
> **W1)** We have an ablation on the strategy used for coarsening (spectrum-preserving, feature-preserving, and random, see Table 12 in Appendix I). For the rebuttal, we also ran a sensitivity analysis on the *rand_lambda* parameter for spectrum-preserving coarsening, whose results are shown in the following table:
>
> ## ManifoldNet Airplane
> |Rand λ|Chamfer Distance|Spectral|Node Degree|Edge Size|
> |-|-|-|-|-|
> |0.4|0.060|0.009|0.23|0.006|
> |0.3|0.070|0.013|0.23|0.009|
> |0.2|0.067|0.012|0.22|0.004|
> |0.1|0.048|0.010|0.22|0.004|
> |0.0|0.058|0.110|0.36|0.009|
>
> ## Planar Graphs
> |Rand λ|Valid Planar (%)|Spectral|Node Degree|Ratio|
> |-|-|-|-|-|
> |0.4|96.9|0.010|0.001|2.3|
> |0.3|96.7|0.008|0.000|2.2|
> |0.2|96.9|0.010|0.000|4.6|
> |0.1|93.8|0.011|0.001|7.8|
> |0.0|100.0|0.008|0.001|2.7|
>
> This shows that the method is relatively robust to the choice of hyperparameters.
>
> **W2)** We agree that hierarchical methods are not required for the generation of small graphs. We included results in small graphs for completeness, but this is not the main target application of our method. This is true in general for the previous coarsening/expansion methodologies [1, 2]. That being said, our method remains competitive, showing no degradation in performance. Its benefits become evident on large graphs and hypergraphs, where each component improves results (see ablations in Appendix I). This demonstrates the versatility of our approach, as it performs well across both small and large-scale settings without compromise.
>
> In the following table, we compare the sampling time of our method and DiGress (a flat model with quadratic complexity) for generating a single planar graph at varying sizes:
> |Graph size|Digress (s)|Ours (s)|
> |-|-|-|
> |100|20.24|99.95|
> |150|43.69|108.72|
> |200|67.14|117.48|
> |300|94.13|133.63|
> |500|161.39|156.89|
> |700|292.37|189.72|
> |900|466.67|209.62|
> |1100|681.87|225.62|
> |1300|921.12|252.07|
> |1500|1213.03|265.10|
> |1700|1507.54|294.82|
>
> The advantage of the $O(n \log n)$ complexity of hierarchical methods becomes clear as graph size increases, which is reflected in the results above.
>
>
>
> Please also note that flat hypergraph models cannot scale: the number of potential hyperedges grows as 2^n, making them practical only for extremely small hypergraphs.
>
> **Q1, Limitations)** Using standard discrete diffusion is challenging in our setting because coarsening requires “intermediate” values between categories when merging nodes, which discrete diffusion does not naturally provide.
>
> A possible approach is to replace the feature flow-matching with a categorical flow-matching operating on the simplex. In this setup:
> - Coarsening remains unchanged, since merging features on the simplex stays on the simplex. Each cluster’s feature naturally represents the fraction of nodes in each category.
> - At each scale, the model refines features in the simplex until clusters reduce to a single node, at which point the feature becomes a one-hot vector.
>
> We leave this to future work.
>
> We hope the responses have addressed your concerns, and we will modify the paper accordingly in our final version. If you have further suggestions, we are also happy to integrate them.
>
> ---
>
> [1] Bergmeister et al., "Efficient and Scalable Graph Generation," ICLR 2024.
>
> [2] Gailhard et al., "HYGENE: A Diffusion-based Hypergraph Generation Method," AAAI 2025.
>
> [3] Vignac et al., “DiGress: Discrete Denoising Diffusion for Graph Generation,” ICLR 2023.

---

> > ### Author Rebuttal · Reviewer_oSXS · 2026-04-03
> >
> > The authors addressed my concerns and I will keep my positive score.

---

### Official Review · Reviewer_aHjv · 2026-03-12

**Soundness:** 2
**Presentation:** 3
**Significance:** 2
**Originality:** 3
**Overall Recommendation:** 4
**Confidence:** 3

**Summary:**

The paper proposes FAHNES, a hierarchical framework for joint topology and feature generation on graphs and hypergraphs. The method builds multiscale representations via coarsening, then learns the reverse process through expansion + refinement on a bipartite representation, with a hierarchical scale encoding and OT-based coupling. The problem is relevant, especially for featured hypergraphs and 3D structures, and the paper reports competitive results on several datasets.

**Compliance With Llm Reviewing Policy:**

Affirmed.

**Final Justification:**

The follow-up response resolves most of my concerns. The additional clarifications and new evaluation details make the contribution clearer and address the main issues I had raised. I am therefore willing to increase my score to *4 (Weak Accept)*.

**Key Questions For Authors:**

1. Please clarify the exact flow-matching objective used for topology, split variables, and feature variables. What are the source and target states at each scale?
2. Please provide a clean sampling algorithm in the main paper: what is sampled, what is predicted, and what is deterministic during generation?
3. Can you provide runtime and memory comparisons versus baselines, including the cost of OT coupling, since scalability is one of the main claims?
4. For featured 3D generation, can you report diversity/coverage-oriented metrics in addition to Chamfer-based proximity to the training set?
5. Please clarify what you see as the single main technical novelty beyond combining existing hierarchical generation, flow matching, and OT-based alignment.

**Limitations:**

Yes

**Strengths And Weaknesses:**

**Strengths:**

The paper targets an interesting and underexplored setting: hierarchical generation of featured graphs/hypergraphs, which is broader than prior unfeatured hierarchical methods. The multiscale setup is reasonably motivated, and the proposed scale encoding appears to matter empirically. The experiments cover multiple domains, including hypergraphs, graphs, meshes, and point clouds, which gives the paper decent breadth.

**Weaknesses:**

1. The paper is framed as a generative modeling contribution, but the main body is much clearer on coarsening/setup than on the actual generative mechanism. The key expansion/refinement pipeline is harder to follow than it should be.
 2. The flow-matching formulation is not explained with enough precision in the main paper. The manuscript says the model learns expansion/refinement via flow matching, but the actual state definition, interpolation, and variable-wise objectives are not made explicit enough. The probabilistic rationale is pushed to the appendix.
 3. Relatedly, the sampling story is not sufficiently clean. For additional clarity, a simple main-paper algorithm is required which answers: what is sampled at each scale, what is deterministic, and how topology/features are jointly updated during expansion and refinement.
4. In terms of novelty, much of the machinery seems compositional: hierarchical coarsen-then-expand, flow matching, local refinement backbone, and OT coupling are largely adapted from prior ingredients. The most concrete new pieces seem to be the hierarchical scale encoding and the way coupling is adapted in this setting, which makes the contribution feel more incremental than the framing suggests.
 5. The experimental rigor is not fully convincing for an ICML paper:

5.1 the scalability claim is central, but there are no runtime/memory scaling plots, only complexity discussion and some OOM-style evidence;

5.2 for featured 3D generation, evaluation seems too limited if based mainly on Chamfer distance to training samples, which does not say enough about diversity/coverage/memorization;

5.3 the paper does not clearly dominate strong baselines in all settings, so the “state-of-the-art” positioning should be stated more carefully.

---

> ### Author Rebuttal · Authors · 2026-03-30
>
> We thank the reviewer for the thoughtful comments. We will incorporate these points into the potential camera-ready version.
>
> **W1)** The expansion/refinement step is simply the inverse of coarsening. As described in Section 3.4, our approach learns the distribution of inputs to this expansion/refinement process, effectively inverting the algorithmically computed coarsening sequences.
>
> **W2, Q1)** All details of our flow-matching formulation are fully described in Appendix F.1. In short, we follow a standard endpoint flow-matching framework, transporting samples from a prior $p_0$​ to a target $p_1$​ at each scale (see lines 1280-1295). Specifically:
> - **Node and hyperedge expansion**:  target $p_1​$ is −1 or 1 (*no expansion* / *expansion*, see lines 1232-1234).
> - **Edge selection**: target is −1 or 1 (selection or removal, see line 1232).
> - **Scale encoding fractions**: target is the true fraction of each child node (see lines 1236-1239).
> - **Node and hyperedge features**: target is the true feature (see lines 1240-1242).
>
> For scale encoding fractions, $p_0​$ is Dirichlet. For the rest, $p_0​$ is Gaussian.
>
> The total loss is the sum of the mean squared error for all components (see 1211-1224).
>
> **W3, Q2)** We detail the full sampling procedure in Algorithm 5 (Appendix G).
> Concretely, at each scale, we follow prior hierarchical methods [1, 2] and first sample a reduction fraction (see line 9 of Algorithm 5), determining the size of the next scale (see line 10).
> Conditioned on this, the model samples all inputs required for expansion (see line 12):
> - number of expansions per node and hyperedge,
> - edge selection or removal,
> - scale encoding fractions, and
> - refined features.
>
> These are then passed to the deterministic expansion operator defined in Section 3.3 (see line 16). This process is repeated until the desired graph size is obtained. That being said, lines 250 - 262 of Section 3.4 already gave a summarized version of the sampling pipeline.
>
> **W4, Q5)** Please refer to the answer to **W1)** of reviewer **2Way**.
>
> **W5)** We report the mean and standard deviation over 5 independent runs for all experiments. Our evaluation spans **four modalities** across both small-, medium-, and large-scale settings (up to a thousand nodes), which substantially broadens the scope compared to prior work. We also include **comprehensive ablation studies** covering all key components of our method, as well as an ablation on the coarsening strategy (see Table 12 in Appendix I).
>
> Overall, this evaluation is significantly more extensive than most prior graph generation works [3, 4], which typically focus on a single modality (molecules), small graphs, and a limited set of datasets.
>
> **W5.1, Q3)** Please refer to the answer to **W2)** of reviewer **oSXS** for a comparison of the sampling time between our method and DiGress.
> Our OT-coupling component has a 4% computational overhead.
>
> **W5.2, Q4)** The Chamfer distance to the nearest training sample already accounts for memorization. We also include complementary metrics related to topology in Table 7 of Appendix I, providing information regarding the coverage and diversity of the learned distribution, as they measure a distance between the training and predicted distributions.
>
> Please note that coverage and diversity are not standardly reported metrics [5, 6, 7].
>
> **W5.3)** We disagree with this statement: our method is enabling capabilities that were previously out of reach, while remaining competitive with previous methods in their own abilities.
> In particular, it is, to our knowledge, the first method able to generate featured graphs and hypergraphs at this scale, while previous methods have OOM errors (see our experiments in Tables 3 and 4), which represents the main axis of novelty and advancement.
>
> On small-graph datasets, our method remains fully competitive with specialized baselines (having Ratios of 4.8 vs 4.9, 1.8 vs 1.6 and 2.2 vs 1.6 compared to DeFoG, current SOTA for graph generation, on the SBM, Tree and planar datasets).
>
> We hope the responses have addressed your concerns, and we will modify the paper accordingly in our final version. If you have further suggestions, we are also happy to integrate them.
>
> ---
>
> [1] Efficient and Scalable Graph Generation, ICLR 2024.
>
> [2] HYGENE: A Diffusion-based Hypergraph Generation Method, AAAI 2025.
>
> [3] DeFoG: Discrete Flow Matching for Graph Generation, ICML 2025.
>
> [4] DiGress: Discrete Denoising Diffusion for Graph Generation, ICLR 2023.
>
> [5] PointNSP: Autoregressive 3D Point Cloud Generation with Next-Scale Level-of-Detail Prediction.
>
> [6] LION: Latent Point Diffusion Models for 3D Shape Generation, NeurIPS 2022.
>
> [7] Fast Point Cloud Generation with Straight Flows, CVPR 2023.

---

> > ### Author Rebuttal · Reviewer_aHjv · 2026-04-03
> >
> > 1. The rebuttal explains the flow-matching part clearly, but the content must be in main paper not in appendix.
> > 2. The rebuttal makes the sampling pipeline easier to understand, but a self-contained algorithm for the full generative process is needed.
> > 3. The novelty still feels like a combination of known ingredients rather than one sharply defined new mechanism.
> > 4. The scalability concern is only partly addressed, since the authors give limited runtime discussion and note about 4% OT overhead, but not a full runtime/memory comparison against strong baselines.
> >
> > Needs revision
> >
> > 1. The paper needs a writing/organization revision, because the main body is clearer on coarsening and setup than on the actual generative mechanism.
> > 2. Too many key details are pushed to the appendix/rebuttal, which breaks the flow for the reader and makes the core method feel like a black box.
> > 3. The 3D evaluation concern remains, because the response defends the chosen metrics but does not add the diversity/coverage-style evidence that was requested.
> > 4. The SOTA positioning should be stated more carefully: in Table 2, DeFoG is uniformly strong across the small-graph datasets, while FAHNES shows selective gains rather than clear overall dominance. This affects the generalizability aspect.
> >
> > I thank the authors for providing a detailed rebuttal,but majority of the concerns requires a revamp than a small rebuttal hence I wish to maintain my score

---

> > > ### Author Response · Authors · 2026-04-07
> > >
> > > We thank the reviewer for the follow-up and for acknowledging that the rebuttal clarified the flow-matching formulation and the sampling pipeline.
> > >
> > > For the camera-ready version, we expect to have one additional page. We will use this space to address the main presentation concerns raised in your follow-up, specifically by: (i) moving the key flow-matching definitions from the appendix into the main paper, and (ii) adding a self-contained main-paper algorithm for the full generative procedure. These were the two central clarity requests in your acknowledgement, and we agree that making them explicit in the main body will improve readability.
> > >
> > > More broadly, **we note that the concern about organization is not unanimous across reviewers**: for instance, reviewer 2Way explicitly listed the organization of the manuscript among its strengths. That said, we understand your point that the current version explains coarsening more clearly than the generative mechanism, and we will rebalance the presentation accordingly in the final version.
> > >
> > > Regarding novelty, we respectfully disagree with the characterization of the method as only a compositional combination of known ingredients. Our main contribution is not flow matching in isolation, nor coarsen-expand in isolation, but a hierarchical framework that jointly generates topology and features for graphs and hypergraphs at scales where prior featured methods do not operate. We also respectfully note that the **review does not point to specific prior work or references that would provide a concrete basis for the novelty concern**, for example, a method proposing a closely related approach. Without such comparison points, it is difficult for us to address this concern more objectively and precisely beyond the clarifications already provided in our rebuttal.
> > >
> > > For Reason 4), we already have this comparison in the paper. Please see the OOM for the datasets *Airplane Point Clouds* and *Bench Point Clouds* in Table 3 for the baselines DiGress and DeFoG, which are the state of the art. The computational complexity analysis already predicts this.
> > >
> > > For Revision 3), we reran the experiments. The following tables summarize the results for the asked metrics:
> > > ## ManifoldNet Airplanes
> > > | Method           | ChamferMMD        | ChamferDiversity  | ChamferCoverage   |
> > > |------------------|------------------|-------------------|-------------------|
> > > | Ours             | **0.0685**           | 0.0845            | **0.3910**            |
> > > | Flow Matching    | 0.0750 ± 0.0028  | 0.0625 ± 0.0051   | 0.2087 ± 0.1007   |
> > > | Wasserstein GAN  | 0.1294 ± 0.0181  | 0.0369 ± 0.0163   | 0.1043 ± 0.0348   |
> > > | VAE              | 0.1850 ± 0.0423  | **0.1907** ± **0.1299**   | 0.2693 ± 0.1172   |
> > >
> > > ## ManifoldNet Benches
> > > | Method           | ChamferMMD        | ChamferDiversity  | ChamferCoverage   |
> > > |------------------|------------------|-------------------|-------------------|
> > > | Ours             | **0.1030**           | **0.2130**            | **0.5300**            |
> > > | Flow Matching    | 0.1120 ± 0.0042  | 0.1117 ± 0.0073   | 0.3067 ± 0.0249   |
> > > | Wasserstein GAN  | 0.1442 ± 0.0100  | 0.0633 ± 0.0128   | 0.1733 ± 0.0327   |
> > > | VAE              | 0.2104 ± 0.0501  | 0.1950 ± 0.1074   | 0.2751 ± 0.0988   |
> > >
> > > *ChamferCoverage* measures the fraction of reference samples that are the nearest neighbor of at least one generated sample, with higher values indicating better coverage. *ChamferDiversity* is the average pairwise Chamfer distance between generated samples, reflecting their diversity (higher is better). *ChamferMMD* is the average distance from each reference sample to its nearest generated sample, measuring fidelity to the reference distribution (lower is better).
> > >
> > > We will add those metrics in the camera-ready version.
> > >
> > > For Revision 4),  please note that our method is competitive — having close or even better results than DeFoG on small graphs, notably achieving perfect generation of valid tree graphs, which is a known strength of hierarchical methods, while DeFoG had 96.5% of valid tree graphs — while being able to scale to much larger graphs. Indeed, DeFoG has a complexity of $O(n^2)$, whereas our method has complexity $O(n \log(n))$. SparseDiff [8], a follow-up work to DeFoG improving its scalability, reports similar results to us on the protein dataset but retains quadratic complexity.
> > >
> > > -----
> > > [8] SparseDiff: Sparse Discrete Diffusion for Scalable Graph Generation, TMLR 2025

---

### Official Review · Reviewer_2Way · 2026-03-13

**Soundness:** 3
**Presentation:** 3
**Significance:** 3
**Originality:** 3
**Overall Recommendation:** 4
**Confidence:** 3

**Summary:**

This paper introduces FAHNES, a hierarchical flow-matching framework for generating hypergraphs with node features.
The method extends HYGENE by incorporating node features through budget-weighted aggregation in a multi-scale coarsening-expansion process. The authors claim significant efficiency improvements over diffusion-based methods while achieving state-of-the-art generation quality on synthetic hypergraphs, 3D meshes, and point clouds.

**Compliance With Llm Reviewing Policy:**

Affirmed.

**Final Justification:**

My original concerns are about the contributions of the work and hypergraph sensitivities. The authors have addressed my concerns; therefore, I incline to maintain my positive score.

**Key Questions For Authors:**

See above

**Limitations:**

Yes

**Strengths And Weaknesses:**

Strengths

1. The manuscript is well organized, and the audience can capture the key ideas in the paper
2. The experiments are comprehensive, the ablations are thorough, and the complexity discussion provides useful insight into scalability.

Weakness

1. The paper is primarily an engineering contribution combining existing techniques such as flow-matching, bipartite coarsening, and PPGN. The authors should clarify the key contributions of the simple combination.
2. The proposed method contains several hyperparameters. The hyperparameter sensitivity analysis or justification is recommended to further strengthen the claims.

---

> ### Author Rebuttal · Authors · 2026-03-30
>
> We thank the reviewer for the thoughtful comments. We will incorporate these points into the potential camera-ready version.
>
> **W1)** Our main contribution is proposing a method for generating both topology and features for graphs and hypergraphs at scale. This problem is non-trivial and requires several conceptual and technical contributions beyond a simple combination of existing components. As stated in lines 63-76, we introduce:
>
> - **Feature-aware hierarchical generation.**
> We extend the coarsen–expand paradigm to jointly handle structure and features. This requires non-trivial design choices as the coarsen-expand paradigm was tailored for a discrete topology space and not made for continuous features. We justify these choices both theoretically and empirically (see Proposition 1 and our ablation in Table 12). Previous coarsening-expansion strategies were designed exclusively for topology and could not be applied to continuous features.
>
> - **A novel scale encoding for hierarchical methods.**
> A central challenge in hierarchical graph generation is the inhomogeneity of scale across the graph: different regions may grow at different rates, which can severely degrade generation quality (as shown in our ablations in Appendix I.2) and even destroy the training signal (Proposition 4). To address this, we introduce a new scale encoding framework specifically tailored to graphs and hypergraphs. Unlike existing approaches, it enables fully local decision-making: the model does not require any global information (e.g., the total number of nodes or the current stage of generation, which were fed as inputs in [1, 2]), which greatly improves performance (see our ablation in Table 5). To the best of our knowledge, such a mechanism does not exist in prior work and constitutes a key conceptual contribution.
>
> - **A lightweight and novel solution to permutation invariance via graph OT coupling.**
> We propose a new OT-based coupling strategy adapted to graphs. While inspired by OT coupling in the image domain, our setting is fundamentally different: image methods reorder entire samples, whereas we reorder nodes within a sample, which raises distinct theoretical challenges. Our approach is not a direct adaptation; it is supported by new theoretical results (Propositions 2 and 3) and, to our knowledge, represents the first application of such a technique to graph generation.
>
> - **Scalable generation of featured graphs, hypergraphs, and geometric data.**
> Our method is the first to jointly generate topology and features for medium- and large-scale graphs, and marks the first application of featured graph generative models for 3D meshes and point clouds. Existing graph generative models [4, 5] typically suffer from quadratic complexity and do not scale to this regime, as indicated by the OOM errors for large datasets (see Tables 2 and 3). Please also note that these methods are typically applied to small graphs and do not report results for large graphs. We also compare the sampling time between our method and DiGress [4], a representative flat method, in the answer to weakness 2 of reviewer **oSXS**.
>
> We agree that flow-matching is just a building block and not a contribution. That being said, we did not use it out of the box but tailored it to our needs: for features lying on the simplex, we used a tailored flow-matching framework for simplex outputs [3], and we used graph inpainting techniques to alleviate the complexity of the learning problem.
>
> **W2)** Our method contains several hyperparameters (*red_frac_min*, *red_frac_max*, *preserved_eig_size*, *rand_lambda*) for which we reuse most of the values of previous works [1, 2]. The main hyperparameter controlling the coarsening sequences is *rand_lambda*, for which we ran a sensitivity analysis on the *planar* and *ManifoldNetAirplane* datasets for the rebuttals. The results are shown in the following tables:
>
> ## ManifoldNet Airplane
> |Rand λ|Chamfer Distance|Spectral|Node Degree|Edge Size|
> |-|-|-|-|-|
> |0.4|0.060|0.009|0.23|0.006|
> |0.3|0.070|0.013|0.23|0.009|
> |0.2|0.067|0.012|0.22|0.004|
> |0.1|0.048|0.010|0.22|0.004|
> |0.0|0.058|0.110|0.36|0.009|
>
> ## Planar Graphs
> |Rand λ|Valid Planar (%)|Spectral|Node Degree|Ratio|
> |-|-|-|-|-|
> |0.4|96.9|0.010|0.001|2.3|
> |0.3|96.7|0.008|0.000|2.2|
> |0.2|96.9|0.010|0.000|4.6|
> |0.1|93.8|0.011|0.001|7.8|
> |0.0|100.0|0.008|0.001|2.7|
>
> This shows that the method is relatively robust to the choice of hyperparameters.
>
> We hope the responses have addressed your concerns, and we will modify the paper accordingly in our final version. If you have further suggestions, we are also happy to integrate them.
>
> ---
>
> [1] Efficient and Scalable Graph Generation, ICLR 2024.
>
> [2] HYGENE: A Diffusion-based Hypergraph Generation Method, AAAI 2025.
>
> [3] Dirichlet Flow Matching with Applications to DNA Sequence Design, ICML 2024.
>
> [4] DiGress: Discrete Denoising Diffusion for Graph Generation, ICLR 2023
>
> [5] DeFoG: Discrete Flow Matching for Graph Generation, ICML 2025

---

> > ### Author Rebuttal · Reviewer_2Way · 2026-04-02
> >
> > I thank the authors for the rebuttal. The authors' response has addressed my concerns. I incline to maintain my positive scores.

---

### Decision · Program_Chairs · 2026-04-30

**Decision:**

Accept (regular)

**Comment:**

The paper introduces FAHNES, a hierarchical flow-matching framework for generating hypergraphs with node features. Strengths include clear presentation, comprehensive experiments, and strong versatility across graphs and hypergraphs. Weaknesses include limited conceptual novelty, insufficient clarity of key details in the main paper, and incomplete evaluation. Overall, the technical contributions are solid.